# Judging with Confidence: Calibrating Autoraters to Preference Distributions

## Abstract

The alignment of large language models (LLMs) with human values increasingly relies on using other LLMs as automated judges, or "autoraters". However, their reliability is limited by a foundational issue: they are trained on discrete preference labels, forcing a single ground truth onto tasks that are often subjective, ambiguous, or nuanced. We argue that a reliable autorater must learn to model the full distribution of preferences defined by a target population. In this paper, we propose a general framework for calibrating probabilistic autoraters to any given preference distribution. We formalize the problem and present two learning methods tailored to different data conditions: 1) a direct supervised fine-tuning for dense, probabilistic labels, and 2) a reinforcement learning approach for sparse, binary labels. Our empirical results show that finetuning autoraters with a distribution-matching objective leads to verbalized probability predictions that are better aligned with the target preference distribution, with improved calibration and significantly lower positional bias, all while preserving performance on objective tasks.

## 1 Introduction

The alignment of large language models (LLMs) with human values (Ouyang et al., 2022) increasingly relies on using other powerful LLMs as automated judges, or "autoraters", to score model responses. This LLM-as-a-Judge paradigm (Zheng et al., 2023) is now a cornerstone of evaluating and developing safer AI systems, particularly through methods such as reinforcement learning from AI feedback (RLAIF) (Bai et al., 2022).

Currently, autoraters are typically trained on discrete preference labels (Wang et al., 2024b; Kim et al., 2024b; Li et al., 2024), which leads to a fundamental limitation: (collective) human judgment does not correspond to a single label, but rather a distribution (Pavlick & Kwiatkowski, 2019; Nie et al., 2020), especially in complex situations that involve uncertainty or balancing multiple criteria (Arora et al., 2025). Even among qualified annotators, disagreement is common, not simply due to noise but because of systematic differences in how individuals define problems, interpret evidence, or apply values and decision strategies (Mumpower & Stewart, 1996). Current autoraters are trained with a mode-seeking objective that collapses this rich distributional information into a single verdict (e.g., the majority label), which discards crucial uncertainty signals and erases minority viewpoints by construction.

We argue that for an autorater to be reliable, it must be calibrated to model the full distribution of human preferences. An ideal judge should recognize when a topic is contentious (e.g., a 50/50 split), when a preference is clear but not unanimous (e.g., 80/20), and when a judgment is objectively certain. Modeling this distribution is essential for effective risk management, fairness, and building robust alignment systems.

This paper introduces a general and scalable framework for calibrating autoraters' verbal probability predictions to any target preference distribution, while preserving their ability to generate natural-language reasoning traces. We introduce two finetuning methods, each tailored to a set of different data conditions. First, when dense, probabilistic labels from multiple annotators are available, we use direct supervised finetuning (SFT). Second, when only sparse, binary labels are available, we employ a reinforcement learning (RL) approach with rewards based on proper scoring rules, without requiring dense probabilistic annotations during data collection. Our empirical

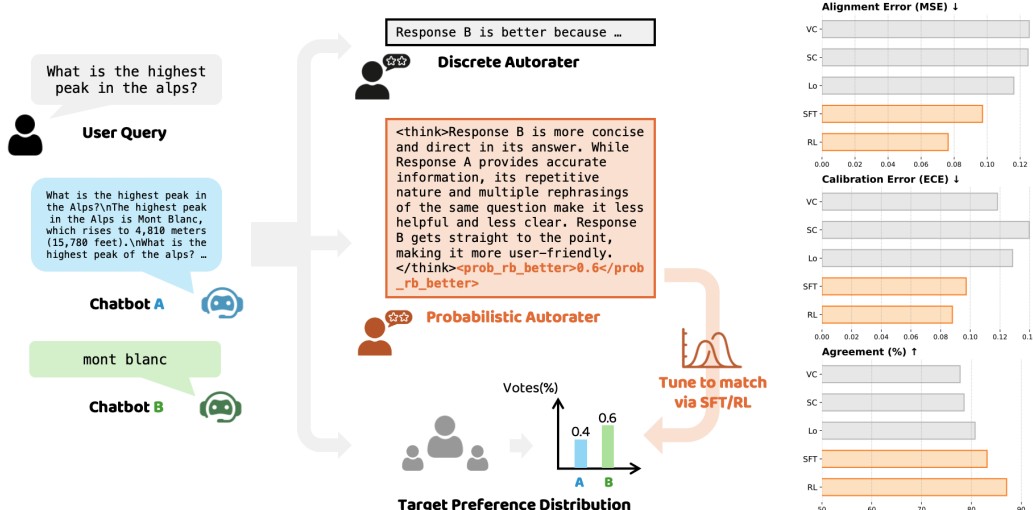

**Figure 1: Overview of discrete vs. probabilistic autoraters**. Left: Given a user query and two candidate responses, a **discrete** autorater returns a single preference (e.g., "B is better"), collapsing annotator variability. A **probabilistic** autorater predicts the *full* preference distribution and is finetuned via SFT/RL to match the target preference distribution. Right: Our finetuned autorater vs. zero-shot probabilistic conversions of discrete autoraters, including Verbalized Confidence (VC), Self-Consistency (SC), and Logits (Lo), evaluated using Gemma-2-9B on JudgeLM *val* set. Alignment error is measured by MSE, calibration error by ECE, and agreement by percentage.

results validate this distribution-matching objective. Autoraters finetuned with our methods show significant improvements in performance, calibration, and reliability. Notably, our methods achieve an 18-51% reduction in Mean Squared Error (MSE), a 4-45% reduction in Expected Calibration Error (ECE), and a 7-81% gain in consistency against positional bias. Our findings offer practical guidance on annotation strategy: for a fixed budget, RL with many sparse, binary labels is more data-efficient than SFT with fewer dense, probabilistic labels, highlighting the benefits of prompt diversity. Our method also enhances alignment with human judgment on out-of-distribution tasks. On the PandaLM dataset, our finetuned Gemma-2-9B model achieves 73.17% agreement with human annotations, outperforming all baselines including GPT-4. Moreover, this improved calibration on subjective tasks does not compromise performance on objective ones, as the same model achieves an overall accuracy of 46.57% on JudgeBench, on par with Gemini-1.5-pro.

## 2 A PROBABILISTIC FRAMEWORK FOR CALIBRATING AUTORATERS

### 2.1 PROBLEM FORMULATION

We consider the scenario of pairwise judgements where an input $\boldsymbol{X}$ specifies a prompt that is associated with two responses $(A, B)$. The population's ground-truth preference is modeled as a Bernoulli random variable

$$Y \in \{0, 1\}, \qquad Y \mid \boldsymbol{X} = \boldsymbol{x} \sim \text{Bernoulli}(p^*(\boldsymbol{x})),$$

where $Y=1$ indicates $B \succ A$ (i.e., $B$ is preferred to $A$), and $p^*(\boldsymbol{x}) = \Pr[Y=1 \mid \boldsymbol{X}=\boldsymbol{x}]$ is the (unknown) preference distribution for the pair $(A, B)$. Let $h$ index a human annotator drawn from the population $p(h)$, then conceptually $p^*(\boldsymbol{x})$ represents the true population-level human preference $p^*(\boldsymbol{x}) = \Pr_{h \sim p(h)}[B \succ A | \boldsymbol{x}, h]$, i.e., the probability that a randomly chosen annotator would prefer $B$ given the context $\boldsymbol{x}$.

**Discrete Autorater.** An autorater is a language model (LM) prompted to act as a judge. In the *discrete* setting, the LM produces a single decision (e.g., via greedy decoding)

$$d_{\boldsymbol{\theta}}(\boldsymbol{x}) \in \{0, 1\},$$

or an uncalibrated scalar margin $m_{\boldsymbol{\theta}}(\boldsymbol{x}) \in \mathbb{R}$ (e.g., a parsed rubric rating or a logit difference), with decision $\mathbb{1}\{m_{\boldsymbol{\theta}}(\boldsymbol{x}) \geq 0\}$. Such outputs collapse the rater distribution at $\boldsymbol{x}$ to a point estimate (typically the majority choice).

**Probabilistic Autorater.** A *probabilistic* judge instead predicts the full preference distribution through its Bernoulli parameter,

$$p_{\boldsymbol{\theta}}(\boldsymbol{x}) \in [0, 1] \approx p^*(\boldsymbol{x}).$$

A definitive decision can be recovered by thresholding $p_{\boldsymbol{\theta}}(\boldsymbol{x})$ if needed, but the primary output is the (conditional) probability itself, which is optimized to match the population preference rather than merely to choose a label. Crucially, this formulation does not rely on the Bradley-Terry assumption (Bradley & Terry, 1952) typically seen in reward modeling, thereby allowing richer representations of uncertainty. Modeling each pair as a conditionally independent Bernoulli trial also enables capturing valid intransitivities (e.g., Condorcet cycles) and ambiguities inherent in collective human judgment, which are often lost when forcing a strict ranking structure.

## 2.2 BENEFITS OF PROBABILISTIC AUTORATERS

**Informativeness.** In contrast to the mode-seeking behavior of discrete autoraters, probabilistic autoraters are optimized to match the full preference distribution. This probability prediction provides more information for cost-sensitive decision-making by revealing the aleatoric ambiguity within the task. Probabilistic reporting in autoraters also improves fairness and auditability by revealing annotator disagreement, whereas discrete reporting collapses the minority viewpoints.

**Alignment.** Probabilistic autoraters can be better aligned with the target preference distribution. In particular, it is straightforward to show that if $\Pr[0 < p^*(\boldsymbol{x}) < 1] > 0$, then any discrete autorater or any single human annotator who effectively reports a degenerate distribution $d(\boldsymbol{x}) \in \{0, 1\}$ is strictly worse than reporting $p(\boldsymbol{x}) = p^*(\boldsymbol{x})$ under a strictly proper scoring rule.

**Calibration.** Additionally, this distribution-matching objective of probabilistic autoraters implies *calibration*: if $p_{\boldsymbol{\theta}}(\boldsymbol{x}) = p^*(\boldsymbol{x})$ almost surely, then $\mathbb{E}[Y \mid p_{\boldsymbol{\theta}}(\boldsymbol{x}) = c] = c$ for all $c \in [0, 1]$. In practice, as $p_{\boldsymbol{\theta}}$ approaches $p^*$, calibration error (e.g., ECE) shrinks. By contrast, any probability prediction obtained from a discrete autorater *post hoc* (e.g., vote fractions from self-consistency or logits passed through softmax) is not trained to recover $p^*(\boldsymbol{x})$ and thus is generally not calibrated.

## 3 FINETUNING AUTORATERS TO MATCH THE PREFERENCE DISTRIBUTION

We introduce two distribution-matching finetuning paradigms for calibrating the autorater's probabilistic prediction $p_{\boldsymbol{\theta}}$ to the ground truth preference distribution $p^*(\boldsymbol{x})$. We focus on *verbalized* probability because it is (1) more flexible and interpretable than training a dedicated classification head by preserving the model's ability to generate natural-language rationales, and (2) more efficient than sampling-based approaches, as it only requires a single decoding pass.

**Setting 1: Direct Supervised Finetuning with Probabilistic Labels.** When multiple annotations $(\boldsymbol{x}, y^{(1)}, ..., y^{(m)})$ are available for each prompt $\boldsymbol{x}$ (the pair to be judged), we estimate the population preference by the multi-annotator mean $\hat{p}(\boldsymbol{x}) = \frac{1}{m} \sum_{j=1}^{m} y^{(j)} \approx p^*(\boldsymbol{x})$. We then instruction-tune the autorater in a text-to-text fashion: given prompt $\boldsymbol{x}$ (the pair), the target sequence includes optional CoT reasoning and a structured numeric field encoding the probability that B is better. We apply standard autoregressive supervised finetuning (SFT) to maximize the likelihood of the target sequence $\boldsymbol{\tau}_{1:S}$ composed using $\hat{p}(\boldsymbol{x})$:

$$\mathcal{L}_{\text{SFT}} = -\mathbb{E}_{(\boldsymbol{x}, \hat{p})} \Big[ \sum_{t=1}^{S} \log P_{\boldsymbol{\theta}}(\boldsymbol{\tau}_t \mid \boldsymbol{x}, \boldsymbol{\tau}_{1:t-1}) \Big],$$

where $\boldsymbol{\tau}_i$ is the $i$th token of the sequence $\boldsymbol{\tau}$ and $S$ is its sequence length.

In practice, we parse the generated string to recover the numeric form of $p_{\boldsymbol{\theta}}(\boldsymbol{x})$; training encourages the model to produce $p_{\boldsymbol{\theta}}(\boldsymbol{x}) \approx \hat{p}(\boldsymbol{x})$ while retaining the ability to perform free-form reasoning.

**Setting 2: Reinforcement Learning from Binary Labels via Piecewise Proper Rewards.** When only single-sample binary labels $(\boldsymbol{x}_i, y_i)$ are available (e.g., via crowdsourced platforms), we treat the autorater as a sequence policy $\boldsymbol{\pi_\theta}(\boldsymbol{\tau} \mid \boldsymbol{x})$ that produces a token sequence $\boldsymbol{\tau}$ containing a numeric probability $p$. A deterministic parser $g$ maps $\boldsymbol{\tau}$ to either a valid probability prediction in $[0, 1]$ or $\perp$ (unparsable), with the probability of producing a parsable response denoted as $s_{\boldsymbol{\theta}}(\boldsymbol{x})$:

$$g : \mathcal{T} \to [0, 1] \cup \{\perp\}, \quad \boldsymbol{\tau} \mapsto p \text{ or } \perp, \qquad s_{\boldsymbol{\theta}}(\boldsymbol{x}) = \Pr_{\boldsymbol{\tau} \sim \boldsymbol{\pi_\theta}(\cdot|\boldsymbol{x})} [g(\boldsymbol{\tau}) \neq \perp].$$

We use *piecewise* strictly proper scoring rules as rewards. Let $y \in \{0, 1\}$ (1 = B better), we have:

- **Brier reward:**
$$R_{\text{Brier}}(\boldsymbol{\tau}; y) = \begin{cases} 1 - (p - y)^2, & \text{if } g(\boldsymbol{\tau}) = p \in [0, 1], \\ 0, & \text{if } g(\boldsymbol{\tau}) = \perp. \end{cases} \tag{1}$$

- **Logarithmic reward (with clipping):** For numerical stability, we consider a fixed small $\epsilon \in (0, \frac{1}{2})$ and define $p' = \text{clip}(p, \epsilon, 1 - \epsilon)$. Then
$$R_{\text{Log}}(\boldsymbol{\tau}; y) = \begin{cases} y \log p' + (1 - y) \log(1 - p'), & \text{if } g(\boldsymbol{\tau}) = p \in [0, 1], \\ \log \epsilon, & \text{if } g(\boldsymbol{\tau}) = \perp. \end{cases} \tag{2}$$

For either reward $R \in \{R_{\text{Brier}}, R_{\text{Log}}\}$, the goal is to maximize the population objective

$$J_R(\boldsymbol{\theta}) = \mathbb{E}_{(\boldsymbol{x}, y)}\Big[\mathbb{E}_{\boldsymbol{\tau} \sim \boldsymbol{\pi_\theta}(\cdot|\boldsymbol{x})}[R(\tau; y)]\Big]. \tag{3}$$

Compared to the Brier reward, the Log reward heavily penalizes overconfident yet incorrect predictions. Both objectives can be optimized with policy-gradient-based reinforcement learning methods using the parsed numeric probability.

### 3.1 Consistency Analysis

In Setting 1, the multi-annotator mean $\hat{p}(\boldsymbol{x})$ is an unbiased estimate of the true preference distribution $p^*(\boldsymbol{x})$ with variance decreasing as $1/m$ (Appendix B.1, Lemma 2), and thereby provides a high fidelity target for learning.

In Setting 2, at the population level, the optimal autorater policy under either the piecewise Brier reward or the clipped Log reward is (i) *parsable* everywhere (i.e., $s_{\boldsymbol{\theta}}(\boldsymbol{x}) = 1$), (ii) *deterministic* in the numeric value it reports, and (iii) reports the *truthful* probability (i.e., recovers the target preference distribution, up to clipping for the Log reward). The proof is included in Appendix B.2.

**Proposition 1** (Fisher Consistency of Brier and Log Rewards)**.** *Assume the policy class can realize, for each $\boldsymbol{x}$, a deterministic numeric output $p_{\boldsymbol{\theta}}(\boldsymbol{x}) \in [0, 1]$ with $s_{\boldsymbol{\theta}}(\boldsymbol{x}) = 1$. Then any global maximizer of $J_R$ in equation 3 satisfies:*

*(a) **Brier:** $p_{\boldsymbol{\theta}}(\boldsymbol{x}) = p^*(\boldsymbol{x})$ for almost all $\boldsymbol{x}$.*

*(b) **Log (with clipping):** $p'_{\boldsymbol{\theta}}(\boldsymbol{x}) = \text{clip}(p^*(\boldsymbol{x}), \epsilon, 1 - \epsilon)$ for almost all $\boldsymbol{x}$.*

*Moreover, for both rewards, any stochasticity in the reported numeric value or any non-zero density associated with unparsable outputs strictly reduces the expected reward; thus, an optimizer is deterministic and fully parsable for almost every $x$.*

## 4 Experimental Setup

Here, we describe the datasets used for calibration and evaluation, our scalable preference-annotation pipeline, the autorater output format and reward instantiation, and the finetuning setup.

**Calibration Data.** We build on a subset of prompts from the JudgeLM corpus (Zhu et al., 2025), which aggregates instruction-following tasks (e.g., Alpaca-GPT4 (Peng et al., 2023), Dolly-15K (Conover et al., 2023)) paired with responses from 11 open-source LLMs (including LLaMA (Touvron et al., 2023), Alpaca (Taori et al., 2023), Vicuna (Chiang et al., 2023)). The source

corpus contains 105K prompts. To ensure the same total *annotation budget* across the two finetuning paradigms, we construct two calibration splits: (i) SFT uses 5K prompts with 10 annotations each; (ii) RL uses 50K prompts with a single annotation each. We also apply swap augmentation (Li et al., 2024) by duplicating each pair with A/B swapped and the label flipped. For evaluation, we sample 1K prompts from the original validation set, each with 10 annotations to form probabilistic labels.

**Preference Annotation.** Since most existing datasets lack sufficient multi-rater annotation for reliable probability estimates, to evaluate our method at scale, by default, we employ *Gemini-2.5-Flash* (Comanici et al., 2025) as an advanced teacher to generate pairwise preference labels with brief rationales. We set the temperature to 1.0 and condition on a randomly sampled persona (Appendix I) to increase coverage and reduce prompt-induced bias. Across calibration and evaluation splits, this yields ∼110K total annotations. For each comparison instance $\boldsymbol{x}$, we convert $m$ independent teacher votes into a probabilistic target $\hat{p}(\boldsymbol{x}) = \frac{1}{m}\sum_{j=1}^{m} \mathbb{1}\{B \succ A\}$. For the SFT + CoT setting, we additionally elicit teacher reasoning traces as expert demonstrations by conditioning on the obtained $\hat{p}(\boldsymbol{x})$. We note that our framework is source-agnostic, allowing autoraters to be calibrated to *any* target preference distribution independent of the preference source. We empirically validate this generalization capability in Section 5.5.

**Response Format and Reward Instantiation.** The autorater is prompted to compare responses A vs. B and emit (optionally) a chain-of-thought enclosed in `<think>` tags, followed by a single probability within a `<prob_rb_better>` tag representing $p_{\boldsymbol{\theta}}(\boldsymbol{x}) \approx \Pr[B \succ A \mid \boldsymbol{x}]$. Our prompts are provided in Appendix I. In SFT, we maximize the likelihood of the target token sequence that encodes $\hat{p}(\boldsymbol{x})$. In RL, we parse the numeric probability via a deterministic parser $g$ (implemented by rule-based string-matching) and optimize either $R_{\text{Brier}}$ in equation 1 or $R_{\text{Log}}$ in equation 2. Unparsable outputs receive a default reward (0 for Brier; $\log \epsilon$ for Log), which empirically drives the *parsability rate* $s_{\boldsymbol{\theta}}(\boldsymbol{x})$ toward 1. When dense labels are available, the SFT objective provides a low-variance target for $p_{\boldsymbol{\theta}}$; with single-label supervision, the RL objectives remain consistent and, being strictly proper, recover $p^*(\boldsymbol{x})$ in expectation.

**Base Models and Finetuning Protocol.** We finetune the instruction-tuned *Gemma-2-9B* (Gemma Team, 2024) and *Qwen-2.5-7B* (Qwen Team, 2024) models with full-parameter updates for both SFT and RL. For RL, we use GRPO (Shao et al., 2024). For $R_{\text{Log}}$ we set $\epsilon = 10^{-3}$ to avoid degenerate rewards. Full hyperparameters and training details are reported in Appendix C.

## 5 EXPERIMENTAL RESULTS

### 5.1 MAIN EVALUATION

In this section, we empirically demonstrate that our distribution-matching finetuning approaches can lead to better performing and calibrated autoraters.

**Baselines.** We consider the following four types of *zero-shot baselines* that can be immediately applied to any existing discrete autoraters to obtain probabilistic predictions:

(1) *Verbalized Confidence (Tian et al., 2023)*: The autorater is directly prompted to provide a confidence score without intermediate reasoning.

(2) *Verbalized Confidence w/ CoT (Wei et al., 2022)*: The autorater first generates a step-by-step chain-of-thought explanation before providing its confidence score.

(3) *Self-Consistency (Wang et al., 2023)*: The autorater aggregates preferences over $N$ independent CoT samples. The confidence for a response is the fraction of samples that voted for it.

(4) *Logit-based Confidence*: Confidence is derived by applying a softmax function to the model's output logits $z_{\tau_i}$ for the verbalized preference tokens $\tau_i$ ("A" or "B"), i.e., $p(y = i \mid \boldsymbol{x}) = e^{z_{\tau_i}}/\sum_i e^{z_{\tau_i}}$, $i \in \{0, 1\}$.

Additionally, we consider the following *calibration baselines* that extend the logit-based confidence:

(1) *Temperature Scaling (Guo et al., 2017)* is a post-hoc calibration method that rescales pre-softmax logits $\boldsymbol{z}$ by a single scalar temperature $T > 0$ learned on a held-out calibration

**Table 1:** Main experiment results comparing our methods against zero-shot and calibration baselines on two models. We evaluate alignment (MSE), performance (Agreement, F1 Score), and calibration (ECE, Brier).

| Model | Method | Alignment | Performance | | Calibration | |
|---|---|---|---|---|---|---|
| | | $MSE\downarrow$ | $Agr.\uparrow$ | $F1\uparrow$ | $ECE\downarrow$ | $Brier\downarrow$ |
| **Gemma-2-9B** | *Zero-shot Baselines* | | | | | |
| | Verbalized Confidence | 0.1255 | 0.7773 | 0.5260 | 0.1183 | 0.1615 |
| | Verbalized Confidence w/ CoT | 0.1065 | 0.7893 | 0.5345 | 0.0869 | 0.1445 |
| | Self-consistency (N=10) | 0.1248 | 0.7853 | 0.5482 | 0.1397 | 0.1551 |
| | Self-consistency (N=30) | 0.1217 | 0.7921 | 0.5361 | 0.1374 | 0.1514 |
| | Logits | 0.1162 | 0.8074 | 0.5665 | 0.1285 | 0.1416 |
| | *Calibration Baselines* | | | | | |
| | Temperature Scaling | 0.0839 | 0.8074 | 0.5665 | **0.0827** | 0.1224 |
| | Contextual Calibration | 0.1384 | 0.7753 | 0.5226 | 0.1598 | 0.1728 |
| | Batch Calibration | 0.1153 | 0.8104 | 0.5482 | 0.1255 | 0.1406 |
| | *Ours* | | | | | |
| | SFT | 0.0972 | 0.8314 | 0.5623 | 0.0972 | 0.1257 |
| | SFT w/ CoT | 0.1033 | 0.8214 | 0.5575 | 0.1111 | 0.1332 |
| | RL (Brier) | **0.0764** | **0.8706** | **0.5895** | 0.0879 | **0.0946** |
| | RL (Log) | 0.0934 | 0.8545 | 0.5780 | 0.1141 | 0.1173 |
| **Qwen-2.5-7B** | *Zero-shot Baselines* | | | | | |
| | Verbalized Confidence | 0.1823 | 0.6723 | 0.4486 | 0.1846 | 0.2276 |
| | Verbalized Confidence w/ CoT | 0.1571 | 0.7241 | 0.4866 | 0.1693 | 0.1965 |
| | Self-consistency (N=10) | 0.1916 | 0.7091 | 0.4765 | 0.2168 | 0.2314 |
| | Self-consistency (N=30) | 0.1840 | 0.7251 | 0.4861 | 0.2075 | 0.2212 |
| | Logits | 0.1775 | 0.7382 | 0.4982 | 0.2102 | 0.2133 |
| | *Calibration Baselines* | | | | | |
| | Temperature Scaling | 0.1173 | 0.7402 | 0.4982 | 0.1529 | 0.1646 |
| | Contextual Calibration | 0.1551 | 0.7632 | 0.5159 | 0.1888 | 0.1893 |
| | Batch Calibration | 0.1796 | 0.7402 | 0.4978 | 0.2129 | 0.2162 |
| | *Ours* | | | | | |
| | SFT | 0.1143 | 0.8264 | 0.5590 | 0.1341 | 0.1394 |
| | SFT w/ CoT | 0.1033 | 0.8122 | **0.6075** | 0.1095 | 0.1324 |
| | RL (Brier) | **0.0893** | **0.8575** | 0.5804 | **0.1015** | **0.1103** |
| | RL (Log) | 0.1192 | 0.8244 | 0.5580 | 0.1472 | 0.1474 |

set by minimizing negative log likelihood, producing calibrated confidences $\hat{p}_{\text{TS}}(\boldsymbol{y} \mid \boldsymbol{x}) = \text{softmax}(\boldsymbol{z}/T)$.

(2) *Contextual Calibration (Zhao et al., 2021)* is a test-time debiasing method that estimates the prompt-induced prior using a content-free probe (e.g., "N/A"), then corrects predictions by subtracting this bias in logit space (or dividing probabilities): $\hat{p}_{\text{CC}}(\boldsymbol{y} \mid \boldsymbol{x}) = \boldsymbol{W}\boldsymbol{p}(\boldsymbol{y} \mid \boldsymbol{x})$, where $\boldsymbol{W} = \text{diag}(\boldsymbol{p}(\boldsymbol{y} \mid [\text{N/A}]))^{-1}$ makes the content-free prediction uniform and reduces bias.

(3) *Batch Calibration (Zhou et al., 2024)* is a zero-shot, inference-only correction that estimates the contextual bias $\boldsymbol{b}$ from the current test batch $\{\boldsymbol{x}_i\}_{i=0}^{B}$ via $\boldsymbol{b} = \mathbb{E}_{\boldsymbol{x} \sim P(\boldsymbol{x})}\boldsymbol{p}(\boldsymbol{y} \mid \boldsymbol{x}) \approx \frac{1}{B}\sum_{i=1}^{B}\boldsymbol{p}(\boldsymbol{y} \mid \boldsymbol{x}_i)$. Each example is then calibrated by dividing by this bias term (or equivalently, subtracting $\log \boldsymbol{b}$ from logits): $\hat{p}_{\text{BC}}(\boldsymbol{y} \mid \boldsymbol{x}) \propto \boldsymbol{p}(\boldsymbol{y} \mid \boldsymbol{x})/\boldsymbol{b}$. To ensure estimation accuracy, we use the entire test set in our experiments to estimate the bias.

**Metrics.** We assess the following three aspects of the autorater: (1) its *alignment* to the target preference distribution, as measured by the Mean Squared Error (MSE) between the predicted $p_{\boldsymbol{\theta}}(\boldsymbol{x})$ and the high fidelity estimate $\hat{p}(\boldsymbol{x})$ of the true preference distribution $p^*(\boldsymbol{x})$, (2) its *performance*, in terms of agreement (Agr.) with the majority label (i.e., the mode of the target distribution) and the resulting F1 score, and (3) its *calibration*, as measured by Expected Calibration Error (ECE) (Guo et al., 2017) and Brier score. A more detailed discriptions of these metrics are provided in Appendix C.

**Distribution-Matching Tuning Improves Preference Calibration.** As shown in Table 1, our distribution-matching finetuning methods consistently outperform both zero-shot and calibration baselines across all metrics. While prompting strategies like Chain-of-Thought and self-consistency improve upon simple verbalized confidence, they still result in high alignment errors. In contrast, our finetuning approach drastically reduces this error. For instance, RL with a Brier reward achieves an MSE of just $0.0764$ on Gemma-2-9B. This superior alignment translates directly into stronger performance and better calibration. Notably, on Gemma-2-9B, the RL-Brier model attains the highest agreement (0.8706), F1 score (0.5895), and lowest Brier score (0.0946) among all methods. The

**Table 2:** Evaluation of positional bias. We report *Consistency* (higher is better) and expected *Absolute Symmetry Deviation* ($\mathbb{E}[|\Delta_{SD}|]$, lower is better).

| Method | Gemma-2-9B | | Qwen-2.5-7B | |
|---|---|---|---|---|
| | *Consistency* ↑ | *Abs. Dev.* ↓ | *Consistency* ↑ | *Abs. Dev.* ↓ |
| *Zero-shot Baselines* | | | | |
| Verbalized Confidence | 0.7301 | 0.2242 | 0.4964 | 0.3461 |
| Verbalized Confidence w/ CoT | 0.8094 | 0.1709 | 0.6399 | 0.3120 |
| Logits | 0.7963 | 0.1912 | 0.6529 | 0.3388 |
| *Calibration Baselines* | | | | |
| Temperature Scaling | 0.7963 | 0.1239 | 0.6489 | 0.1953 |
| Contextual Calibration | 0.7021 | 0.3202 | 0.7422 | 0.2473 |
| Batch Calibration | 0.7973 | 0.1893 | 0.6549 | 0.2130 |
| *Ours* | | | | |
| SFT | 0.8375 | 0.1875 | 0.8284 | 0.1827 |
| SFT w/ CoT | 0.7803 | 0.2291 | 0.8335 | 0.1654 |
| RL (Brier) | **0.8926** | **0.1026** | **0.9007** | **0.0964** |
| RL (Log) | 0.8776 | 0.1231 | 0.8726 | 0.1259 |

benefits are even more pronounced on Qwen-2.5-7B, where the same model achieves an agreement of $0.8575$—a $12.4\%$ improvement over Contextual Calibration, the best-performing baseline—while simultaneously achieving best calibration, as measured by both ECE and Brier score.

**RL on Binary Labels is More Annotation-Efficient than SFT on Probabilistic Labels.** A key finding is that for a fixed annotation budget, RL is a more annotation-efficient training paradigm than SFT. As seen in Table 1, RL-tuned autoraters, trained on 50K prompts with a single binary label each, consistently outperform their SFT counterparts, which were trained on 5K prompts with 10 aggregated labels each. We attribute this to the benefits of data diversity: the performance boost from seeing a $10\times$ larger set of unique prompts appears to outweigh the advantage of learning from a less noisy, aggregated target on a smaller dataset. Within the RL framework, the Brier reward consistently yields better results than the Log reward. This is likely because the Log reward's heavy penalties for tail miscalibrations can introduce training instability, whereas the Brier reward provides a smoother optimization landscape.

## 5.2 EVALUATION OF POSITIONAL BIAS

LM-based autoraters, even those based on powerful proprietary models such as GPT-4, are known to be susceptible to positional bias (Zheng et al., 2023; Wang et al., 2024a), which causes their final verdict to be dependent on the order of the responses and thus undermines the reliability of their judgment result. To evaluate the positional bias of the probabilistic autoraters, for each prompt $x$, we perform inference twice by swapping the order of the responses to obtain two predicted probabilities: $p_{\text{orig}}$ that predicts $\Pr[B \succ A \mid x]$ and $p_{\text{swap}}$ that predicts $\Pr[A \succ B \mid x]$.

**Metrics.** Following Zheng et al. (2023), we measure *consistency*, i.e., the ratio of cases where the autorater gives consistent verdicts when swapping the order of the two responses. Additionally, we measure *Symmetry Deviation* ($\Delta_{SD}$) as $\Delta_{SD} := p_{\text{orig}} + p_{\text{swap}} - 1$. Ideally, an unbiased autorater should produce $p_{\text{orig}}$ and $p_{\text{swap}}$ that sum to 1, and thus the estimated $\Delta_{SD}$ would approximate 0. A positive $\Delta_{SD}$ indicates bias toward response B, and vice versa. We report the expected *Absolute Symmetry Deviation* across the dataset as $\mathbb{E}[|\Delta_{SD}|] \approx \frac{1}{N} \sum_{i=1}^{N} |p_{\text{orig},i} + p_{\text{swap},i} - 1|$.

**Distribution-Matching Tuning Reduces Positional Bias.** Probabilistic autoraters, like their discrete counterparts, are

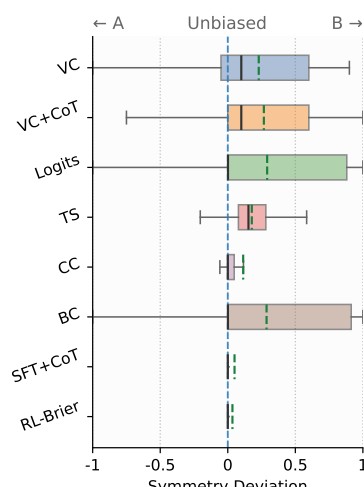

**Figure 2:** Positional bias by method for Qwen-2.5-7B. Each horizontal box shows the distribution of *Symmetry Deviation* ($\Delta_{SD}$): 0 is swap-symmetric, $-1$ indicates bias toward A, and $+1$ toward B. The **black solid** line marks the *median*, while the **green dashed** line marks the *mean*.

susceptible to positional bias, as shown in Table 2. This bias can be severe. For example, the zero-shot verbalized confidence method on Qwen-2.5-7B yields a poor consistency of just 0.4964. While other baselines, including CoT prompting and post-hoc calibration, can mitigate this issue, significant bias remains. For instance, on Qwen-2.5-7B, no single baseline excels at both metrics, with Contextual Calibration achieving the highest consistency (0.7422) and Temperature Scaling achieving the lowest deviation (0.1953). By contrast, our distribution-matching finetuning nearly eliminates this bias. The RL-Brier model, in particular, achieves a consistency of 0.9007 and a near-perfect absolute symmetry deviation of 0.0964. This dramatic improvement is visualized in Figure 2, which shows that baseline methods exhibit heavily skewed deviation distributions, indicating a systematic preference for one response position (in this case, response B). Conversely, our finetuned models center the distribution tightly around zero, demonstrating robust swap-symmetry and verifying their effectiveness at debiasing autoraters.

## 5.3 Out-of-Distribution Evaluation on Human-Annotated Data

To validate our approach against genuine human judgments, we conduct an out-of-distribution evaluation on the PandaLM test set (Wang et al., 2024b) using our autoraters finetuned on JudgeLM subset. This benchmark contains 1K samples, each independently annotated by three human experts. Following the standard protocol for this dataset, we treat the majority vote as the ground truth and report agreement, precision, recall, and F1 score.

**Table 3:** Comparison of autorater performance on the PandaLM (Wang et al., 2024b) test set based on human-annotated data. We report Agreement, Precision, Recall, and F1 Score. Results marked by [*] are reported by Zhu et al. (2025).

| Method | Agreement↑ | Precision↑ | Recall↑ | F1↑ |
|---|---|---|---|---|
| *Zero-shot Baselines* | | | | |
| GPT-3.5[*] | 0.6296 | 0.6195 | 0.6359 | 0.5820 |
| GPT-4[*] | 0.6647 | 0.6620 | 0.6815 | 0.6180 |
| *Finetuned Baselines* | | | | |
| PandaLM-7B[*] | 0.5926 | 0.5728 | 0.5923 | 0.5456 |
| JudgeLM-7B[*] | 0.6507 | 0.6689 | **0.7195** | 0.6192 |
| *Ours (Gemma-2-9B)* | | | | |
| SFT | 0.6856 | 0.7103 | 0.5196 | 0.4998 |
| SFT w/ CoT | 0.7247 | 0.6533 | 0.6166 | 0.6266 |
| RL (Brier) | 0.7317 | 0.6983 | 0.6048 | 0.6220 |
| RL (Log) | **0.7357** | 0.4923 | 0.5487 | 0.5176 |
| *Ours (Qwen-2.5-7B)* | | | | |
| SFT | 0.7027 | 0.4720 | 0.5240 | 0.4947 |
| SFT w/ CoT | 0.7187 | 0.6358 | 0.6522 | **0.6417** |
| RL (Brier) | 0.7297 | **0.8185** | 0.5617 | 0.5564 |
| RL (Log) | 0.7157 | 0.8129 | 0.5454 | 0.5361 |

**Calibrated Autoraters Are Better Aligned with Human Preference.** As shown in Table 3, our models demonstrate superior alignment with human judgments compared to both powerful zero-shot models like GPT-4 and specialized, finetuned judges, including PandaLM-7B (Wang et al., 2024b) and JudgeLM-7B (Zhu et al., 2025). The results are particularly compelling given the data efficiency of our method. For example, our Qwen-2.5-7B model, after SFT with CoT, achieves a state-of-the-art F1 score of 0.6417. This performance surpasses JudgeLM-7B, a model trained on the full 100K JudgeLM training set, i.e., $20\times$ more data than what's used by our SFT model. Even without CoT, our SFT model achieves an agreement of 0.7027, outperforming all baselines, including GPT-4. These findings confirm that our distribution-matching framework is a highly data-efficient method for aligning autoraters with nuanced human preferences.

## 5.4 Out-of-Distribution Evaluation on Objective Tasks

To assess performance on tasks with a single ground truth (i.e., the target preference distribution degenerates to a single point), we evaluate our models on JudgeBench (Tan et al., 2025), a benchmark comprising four objective tasks (Knowledge, Reasoning, Mathematics, and Coding) with binary, verifiable labels. This benchmark allows for comparison against a diverse set of models, including state-of-the-art proprietary APIs such as GPT-4o and Gemini-1.5-pro, multi-agent judges such as ChatEval (Chan et al., 2024), and several specialized finetuned judges, including PandaLM-7B (Wang et al., 2024b), Prometheus2-7B (Kim et al., 2024b), JudgeLM-7B (Zhu et al., 2025), AutoJ-13B (Li et al., 2024), and Skyuwork-Critic-8B (Shiwen et al., 2024). Following the official protocol (Tan et al., 2025), we mitigate positional bias by evaluating each response pair twice, with swapped response order, and aggregating the results to obtain the final verdict.

**Calibrated Autoraters Remain Performant on Objective Tasks.** As shown in Table 4, training our autoraters to model preference distributions does not compromise their performance on objective tasks. Our RL-Brier tuned Gemma-2-9B model, for instance, achieves the *highest* accuracy

of any model on the reasoning task (55.10%). Its overall accuracy of 46.57% surpasses strong baselines like Gemini-1.5-pro and all other finetuned judges except for Skywork-Critic-8B, which was trained on a substantially larger dataset[1]. Our Qwen-2.5-7B model is also highly competitive, achieving an overall accuracy of 44.86%. These results demonstrate that our calibration framework produces versatile probabilistic autoraters that excel at judging subjective tasks without sacrificing their effectiveness on objective, fact-based evaluations.

**Table 4:** Evaluation of autoraters on JudgeBench (Tan et al., 2025) across four objectively labeled tasks: Knowledge, Reasoning, Mathematics, and Coding. We report evaluation accuracy in percentage. Results marked by [*] are reported by Tan et al. (2025).

| Method | Knowledge | Reasoning | Math | Coding | Overall |
|---|---|---|---|---|---|
| *Zero-shot Baselines* | | | | | |
| GPT-4o[*] | 44.16 | 47.96 | 66.07 | **61.90** | 50.86 |
| Gemini-1.5-pro[*] | 45.45 | 44.90 | 53.57 | 28.57 | 44.57 |
| *Multi-Agent Baseline* | | | | | |
| ChatEval[*] | 32.47 | 31.63 | 44.64 | 30.95 | 34.00 |
| *Finetuned Baselines* | | | | | |
| PandaLM-7B[*] | 9.09 | 21.43 | 7.14 | 16.67 | 13.14 |
| Prometheus2-7B[*] | 38.31 | 25.51 | 35.71 | 42.86 | 34.86 |
| JudgeLM-7B[*] | 23.38 | 29.59 | 32.14 | 11.90 | 25.14 |
| AutoJ-13B[*] | 40.26 | 29.59 | 44.64 | 28.57 | 36.57 |
| Skywork-Critic-8B[*] | **51.30** | 54.08 | **73.21** | 33.33 | **53.43** |
| *Ours (RL w/ Brier)* | | | | | |
| Qwen-2.5-7B | 39.61 | 46.94 | 60.71 | 38.10 | 44.86 |
| Gemma-2-9B | 39.61 | **55.10** | 58.93 | 35.71 | 46.57 |

## 5.5 GENERALIZATION ACROSS PREFERENCE SOURCES

We further evaluate the source-agnostic robustness of our framework to ensure it learns valid preference representations independent of the specific preference distribution. To this end, we conducted evaluations on Qwen-2.5-7B using *GPT-5-mini* as an alternative preference source.

As shown in Table 5, we compare three settings: (1) a zero-shot verbalized confidence baseline, (2) our method finetuned directly on GPT-5-mini generated preferences (for testing source independence), and (3) our method finetuned on Gemini-2.5-Flash generated preferences but evaluated on GPT-5-mini (for evaluating cross-preference distribution transferability).

**Source Independence.** Our method successfully calibrates the autorater when finetuned on GPT-5-mini preferences, achieving a Brier score of 0.1321. This confirms that the improvements reported in Section 5.1 are driven by the distribution-matching objective itself, rather than the specific choice of the Gemini preference distribution.

**Cross-Preference Distribution Transferability.** We further evaluated the transferability of our model finetuned on Gemini-2.5-Flash preferences to the unseen GPT-5-mini distribution. Despite a natural disagreement rate of 38.5% between these two preference sources, our RL-tuned model maintains a high agreement of 0.8355 with the GPT preference distribution. This indicates that our method captures fundamental properties of response quality that generalize across distinct preference distributions, rather than overfitting to a specific distribution's artifacts.

**Table 5:** Generalization analysis. We evaluate all models against the GPT-5-mini preference distribution. Our RL method maintains high agreement even when finetuned on a different source (Gemini-2.5-Flash), approaching the performance of a model finetuned directly on the target via SFT.

| Method | Finetuning Source | Alignment (MSE) ↓ | Performance (Agr.) ↑ | Calibration (Brier) ↓ |
|---|---|---|---|---|
| Verbalized Confidence | — | 0.1895 | 0.6824 | 0.2208 |
| Ours (SFT) | GPT-5-mini | 0.1156 | 0.8435 | 0.1321 |
| Ours (RL-Brier Transfer) | Gemini-2.5-Flash | 0.1224 | 0.8355 | 0.1407 |

## 6 RELATED WORK

**Uncertainty of Human Annotations.** There is a growing recognition that human-annotated data is not monolithic. Researchers have highlighted the importance of modeling label ambiguity and disagreement in standard classification tasks (Nie et al., 2020; Baan et al., 2022; Zhou et al., 2022).

---

[1]While the exact size of the training data is not disclosed, the Skywork-Critic-8B model is described as having been finetuned on an array of high-quality datasets, including the Skywork-Reward-Preference dataset (80K samples), the Open-Critic-GPT dataset (55K samples), and other human-annotated and synthetic data.

Most relevant to our work, Elangovan et al. (2025) argue that standard correlation metrics for evaluating LLM judges are insufficient and propose new metrics that stratify data based on human label uncertainty. We take the next step by not only evaluating with respect to this uncertainty, but also proposing methods to directly train models to capture it.

**Calibration of LLMs.** The calibration of LLMs is a well-studied problem. Early work focused on post-hoc calibration methods or prompting strategies to elicit confidence (Tian et al., 2023; Xiong et al., 2024). Other approaches use supervised fine-tuning to teach models to express uncertainty, for example by using a model's own empirical accuracy as a target label (Lin et al., 2022). More recently, reinforcement learning has been used to improve calibration. Xu et al. (2024) use a quadratic reward function with PPO to calibrate a model for question answering. Tao et al. (2024) combine a ranking loss with an order-preserving reward to align a model's outputs. In the context of reward modeling, Leng et al. (2025) address the overconfidence of reward models directly by proposing PPO variants to align quality with verbalized confidence. In concurrent work, Stangel et al. (2025); Damani et al. (2025) propose designing rewards with proper scoring rules to improve confidence calibration on question-answering tasks. Our work differs by focusing specifically on the autorater calibration problem and by proposing a framework grounded in modeling the true distribution of human preferences, rather than a single notion of correctness.

**LLM-as-a-Judge.** The use of powerful large language models as automated evaluators has been explored extensively. This includes the creation of benchmarks (Zheng et al., 2023; Zeng et al., 2024; Tan et al., 2025), analyses of various biases (Wang et al., 2024a; Ye et al., 2024), and methods for training specialized judge models (Wang et al., 2024b; Kim et al., 2024a; Zhu et al., 2025; Li et al., 2024; Saha et al., 2025). Additionally, recent work has also investigated the role of model uncertainty (Xie et al., 2025) and non-transitivity (Xu et al., 2025) in LLM judge evaluations. Our work contributes to this line by addressing a fundamental aspect of judge reliability: its calibration to the inherent subjectivity of the evaluation task. Our work also differs from recent work on Generative Reward Models (Mahan et al., 2024), which use Chain-of-Thought to improve judgment accuracy on complex reasoning tasks such as math and code. In contrast, our framework focuses on aleatoric uncertainty, i.e., calibrating autoraters to match the subjective preference distribution of a population, rather than converging to a single correct reasoning path.

## 7 CONCLUSION

In this paper, we address the core limitation of training autoraters on discrete preference labels, a practice that overlooks the subjective and distributional nature of human judgment. We introduced a general probabilistic framework to calibrate autoraters to model the full preference distribution. Our empirical results show that finetuning with distribution-matching objective leads to autoraters that are better aligned with target preference distributions, with significant improvements in calibration and a substantial reduction in positional bias. By shifting the goal from predicting a single verdict to modeling the spectrum of human opinion, this work contributes to building more reliable, fair, and robust AI alignment systems.

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

## A    EXTENDED RELATED WORK

**Disagreement in Human Judgments.**    Psychology studies have shown that even when individuals are presented with the same evidence, they can rationally arrive at different choices for what is the "best" because of systematic differences in their judgmental processes (Mumpower & Stewart, 1996), including (i) different problem definitions, where disagreement stems from judging different environmental criteria or a confusion between scientific facts and personal value, (ii) different organizing principles, where individuals may apply different cue weights, function forms, or overall biases when integrating the same set of information, as explained by Social Judgment Theory (Brehmer, 1976), and (iii) different mental models about how the evidence was generated. In our context, this explains why human annotators can disagree about LLM outputs even under identical prompts or instructions: they face different value trade-offs (e.g., safety vs. helpfulness) or apply different thresholds for judgment, which may result in distinct yet internally coherent choices of what is preferred.

**Probabilistic Models of Agreement.**    Probabilistic modeling of agreement dates back to the 1950s. Classic models such as the Dawid & Skene (1979) model aim to infer a single "gold standard" label from multiple, often noisy, annotators. Studies (Passonneau & Carpenter, 2014; Paun et al., 2018) have shown that by modeling annotator reliability, these methods can produce high-quality data even from non-expert crowd workers and outperform simpler aggregation techniques such as majority vote. In contrast to this approach, a recent line of work shows that, for many complex and subjective tasks, disagreement is not simply noise but a valid and reproducible signal that reflects legitimate differences in human interpretation (Pavlick & Kwiatkowski, 2019; Nie et al., 2020). Our probabilistic autorater aligns with this view by treating the annotator heterogeneity as the prediction target to capture the full distribution of human judgments.

**Learning from Human Feedback.**    Canonical Reinforcement Learning from Human Feedback (RLHF) (Ouyang et al., 2022) involves learning a reward model from pairwise human preferences, often by fitting a Bradley-Terry model via maximum likelihood estimation. This approach implicitly assumes that heterogeneous feedback from different humans is merely a noisy estimate of a single ground-truth preference. However, a growing body of work (Munos et al., 2024; Ge et al., 2024; Siththaranjan et al., 2024) suggests that, in the context of AI alignment, preference heterogeneity

often reflects legitimate differences in individual values and should be modeled directly rather than averaged away. Our research complements this line of work as our finetuned probabilistic autoraters can be employed in such preference-based RL to better align models with the diversity of human preferences.

# B PROOFS

## B.1 PREFERENCE DISTRIBUTION AND ESTIMATION

Let $h$ index an annotator drawn from a population distribution $p(h)$. For a pairwise input $\boldsymbol{X} = \boldsymbol{x}$, define the annotator-specific preference probability

$$p_h(\boldsymbol{x}) \;=\; \Pr[Y = 1 \mid \boldsymbol{X} = \boldsymbol{x}, h], \qquad Y \in \{0, 1\} \text{ (1 indicates } B \succ A).$$

The population (ground-truth) preference is the annotator-average:

$$p^*(\boldsymbol{x}) \;=\; \mathbb{E}_{h \sim p(h)}[\, p_h(\boldsymbol{x}) \,].$$

A common special case here is when annotators hold *stable* preferences, i.e., where $p_h(\boldsymbol{x}) \in \{0, 1\}$ (each annotator has a fixed judgment per $\boldsymbol{x}$). In that case, $p_h(\boldsymbol{x})$ is the indicator of "$h$ prefers B" and $p^*(\boldsymbol{x})$ is the population fraction preferring B.

Given $m$ i.i.d. labels $\{y^{(j)}\}_{j=1}^m$ collected by sampling annotators $h_j \overset{\text{i.i.d.}}{\sim} p(h)$ and then $y^{(j)} \sim$ Bernoulli($p_{h_j}(\boldsymbol{x})$), the Monte Carlo estimator

$$\hat{p}_m(\boldsymbol{x}) \;=\; \frac{1}{m} \sum_{j=1}^m y^{(j)}$$

is an unbiased estimate of $p^*(\boldsymbol{x})$ with variance decreasing as $1/m$.

**Lemma 2** (Unbiasedness and variance of the multi-annotator estimate). *For any fixed $\boldsymbol{x}$ and i.i.d. sampling as above,*

$$\mathbb{E}[\hat{p}_m(\boldsymbol{x})] \;=\; p^*(\boldsymbol{x}), \qquad \mathrm{Var}[\hat{p}_m(\boldsymbol{x})] \;=\; \frac{p^*(\boldsymbol{x})\big(1 - p^*(\boldsymbol{x})\big)}{m}.$$

*Proof.* By the law of total expectation, $\mathbb{E}[y^{(j)} \mid \boldsymbol{x}] = \mathbb{E}_h[p_h(\boldsymbol{x})] = p^*(\boldsymbol{x})$, so $\mathbb{E}[\hat{p}_m(\boldsymbol{x})] = p^*(\boldsymbol{x})$. Since the $y^{(j)}$ are i.i.d. Bernoulli with mean $p^*(\boldsymbol{x})$ (marginalizing over $h$), $\mathrm{Var}(\hat{p}_m(\boldsymbol{x})) = \mathrm{Var}(y^{(1)})/m = p^*(\boldsymbol{x})(1 - p^*(\boldsymbol{x}))/m$. $\square$

## B.2 PROOF OF PROPOSITION 1

We first recall the setup. For $(\boldsymbol{X}, Y) \sim \mathcal{D}$ with $Y \in \{0, 1\}$ and $Y \mid \boldsymbol{X} = \boldsymbol{x} \sim$ Bernoulli($p^*(\boldsymbol{x})$), the policy $\boldsymbol{\pi_\theta}(\boldsymbol{\tau} \mid \boldsymbol{x})$ emits a token sequence $\boldsymbol{\tau}$ intended to encode a numeric probability. A deterministic parser $g : \mathcal{T} \to [0, 1] \cup \{\perp\}$ returns either a number $p \in [0, 1]$ or the unparsable symbol $\perp$. Let $s_{\boldsymbol{\theta}}(\boldsymbol{x}) = \Pr_{\boldsymbol{\tau} \sim \boldsymbol{\pi_\theta}(\cdot | \boldsymbol{x})}[g(\boldsymbol{\tau}) \neq \perp]$. The piecewise rewards are:

$$R_{\mathrm{Brier}}(\boldsymbol{\tau}; y) = \begin{cases} 1 - (p - y)^2, & g(\boldsymbol{\tau}) = p \in [0, 1], \\ 0, & g(\boldsymbol{\tau}) = \perp, \end{cases}$$

$$R_{\mathrm{Log}}(\boldsymbol{\tau}; y) = \begin{cases} y \log p' + (1 - y) \log(1 - p'), & g(\boldsymbol{\tau}) = p \in [0, 1], \\ \log \epsilon, & g(\boldsymbol{\tau}) = \perp, \end{cases}$$

with $p' = \mathrm{clip}(p, \epsilon, 1 - \epsilon)$ and $\epsilon \in (0, \frac{1}{2})$. The population objective is $J_R(\boldsymbol{\theta}) = \mathbb{E}_{(\boldsymbol{X}, Y)} \mathbb{E}_{\boldsymbol{\tau} \sim \boldsymbol{\pi_\theta}(\cdot | \boldsymbol{X})}[R(\boldsymbol{\tau}; Y)]$.

Fix $\boldsymbol{x}$ and abbreviate $p^* = p^*(\boldsymbol{x})$. All statements below are conditional on $\boldsymbol{X} = \boldsymbol{x}$ and the conclusion holds for almost every $\boldsymbol{x}$ (w.r.t. the marginal of $\boldsymbol{X}$).

This proof utilizes the following observations: (i) for any random variable $Z$ with finite variance and any constant $a$, $\mathbb{E}[(Z - a)^2] = (\mathbb{E}[Z] - a)^2 + \mathrm{Var}(Z)$; (ii) the function $\phi(p) = p^* \log p + (1 - p^*) \log(1 - p)$ is strictly concave on $p \in (\epsilon, 1 - \epsilon)$ with unique maximizer at $p = p^*$ (and at the boundary when $p^* \notin (\epsilon, 1 - \epsilon)$).

**Brier.** Let $P$ denote the random numeric report $g(\boldsymbol{\tau})$ conditional on $g(\boldsymbol{\tau}) \neq \bot$ (so $P$ is defined with probability $s_{\boldsymbol{\theta}}(\boldsymbol{x})$). Then

$$
\begin{aligned}
\mathbb{E}_{\boldsymbol{\tau}, Y}[R_{\mathrm{Brier}}(\boldsymbol{\tau}; Y) \mid \boldsymbol{x}] &= s_{\boldsymbol{\theta}}(\boldsymbol{x}) \, \mathbb{E}_{\boldsymbol{\tau}}\big[\, \mathbb{E}_{Y|\boldsymbol{x}}\big[1 - (P - Y)^2\big]\big] + \big(1 - s_{\boldsymbol{\theta}}(\boldsymbol{x})\big) \cdot 0 \\
&= s_{\boldsymbol{\theta}}(\boldsymbol{x}) \, \mathbb{E}_{\boldsymbol{\tau}}\big[1 - \mathbb{E}_{Y|\boldsymbol{x}}\big[(Y - P)^2\big]\big] \\
&= s_{\boldsymbol{\theta}}(\boldsymbol{x}) \, \mathbb{E}_{\boldsymbol{\tau}}\big[1 - \big(\mathbb{E}_{Y|\boldsymbol{x}}[Y] - P\big)^2 - \mathrm{Var}_{Y|\boldsymbol{x}}(Y)\big] \qquad \text{applying (i)} \\
&= s_{\boldsymbol{\theta}}(\boldsymbol{x}) \, \mathbb{E}_{\boldsymbol{\tau}}\big[1 - (p^* - P)^2 - p^*(1 - p^*)\big] \\
&= s_{\boldsymbol{\theta}}(\boldsymbol{x}) \left(1 - \mathbb{E}_{\boldsymbol{\tau}}\big[(P - p^*)^2\big] - p^*(1 - p^*)\right) \\
&= s_{\boldsymbol{\theta}}(\boldsymbol{x}) \left(1 - \big[(\mathbb{E}_{\boldsymbol{\tau}}[P] - p^*)^2 + \mathrm{Var}_{\boldsymbol{\tau}}(P)\big] - p^*(1 - p^*)\right). \quad \text{applying (i)}
\end{aligned}
$$

For fixed $s_{\boldsymbol{\theta}}(\boldsymbol{x})$ this is maximized by setting $\mathrm{Var}(P) = 0$ (deterministic numeric output) and $\mathbb{E}[P] = p^*$ (truthful reporting). Moreover, since $1 - p^*(1 - p^*) > 0$, the expectation is positive when $\mathrm{Var}(P) = 0$ and $\mathbb{E}[P] = p^*$. Increasing $s_{\boldsymbol{\theta}}(\boldsymbol{x})$ strictly increases the expectation; hence an optimizer satisfies $s_{\boldsymbol{\theta}}(\boldsymbol{x}) = 1$. Therefore, at any global maximizer, $P$ is almost surely constant and equals $p^*$, i.e., $p_{\boldsymbol{\theta}}(\boldsymbol{x}) = p^*$.

**Log with clipping.** Write $P'$ for the clipped numeric report when parsable. Then

$$
\begin{aligned}
\mathbb{E}_{\boldsymbol{\tau}, Y}[R_{\mathrm{Log}}(\boldsymbol{\tau}; Y) \mid \boldsymbol{x}] &= s_{\boldsymbol{\theta}}(\boldsymbol{x}) \, \mathbb{E}_{\boldsymbol{\tau}}\big[\, \mathbb{E}_{Y|\boldsymbol{x}}\big[Y \log P' + (1 - Y) \log(1 - P')\big]\big] + (1 - s_{\boldsymbol{\theta}}(\boldsymbol{x})) \log \epsilon \\
&= s_{\boldsymbol{\theta}}(\boldsymbol{x}) \, \mathbb{E}_{\boldsymbol{\tau}}\big[p^* \log P' + (1 - p^*) \log(1 - P')\big] + (1 - s_{\boldsymbol{\theta}}(\boldsymbol{x})) \log \epsilon \\
&= s_{\boldsymbol{\theta}}(\boldsymbol{x}) \, \mathbb{E}_{\boldsymbol{\tau}}[\phi(P')] + (1 - s_{\boldsymbol{\theta}}(\boldsymbol{x})) \log \epsilon.
\end{aligned}
$$

By strict concavity of $\phi$ and Jensen's inequality, $\mathbb{E}[\phi(P')] \leq \phi(\mathbb{E}[P'])$ with equality iff $P'$ is almost surely constant (deterministic numeric output). The maximizer over $P' \in [\epsilon, 1 - \epsilon]$ is uniquely $P' \equiv \mathrm{clip}(p^*, \epsilon, 1 - \epsilon)$ (truthful reporting). Finally, at this maximizer $\phi\big(\mathrm{clip}(p^*, \epsilon, 1 - \epsilon)\big) > \log \epsilon$, so allocating any mass to unparsable outputs (which yields $\log \epsilon$) strictly reduces the expectation; hence $s_{\boldsymbol{\theta}}(\boldsymbol{x}) = 1$. Therefore, the optimal report is deterministic and equals the clipped truth $\mathrm{clip}(p^*, \epsilon, 1 - \epsilon)$.

Combining the two cases completes the proof. $\qquad\square$

## C  IMPLEMENTATION DETAILS

### C.1  EVALUATION DETAILS

We measure the probabilistic autorater's performance by comparing its judgment to human annotation or discretized probabilistic labels annotated by Gemini-2.5-Flash. Following prior work (Wang et al., 2024b; Zhu et al., 2025), we formulate the pairwise judgment task as a three-class classification problem with labels $A \succ B$, Tie, and $B \succ A$. Let $TP_i$, $FP_i$, and $FN_i$ denote the true positives, false positives, and false negatives for class $i$, respectively. We report macro-averaged metrics by computing each score per class and then averaging over all $C$ classes:

$$
\mathrm{Precision}_{\mathrm{macro}} = \frac{1}{C} \sum_{i=1}^{C} \frac{TP_i}{TP_i + FP_i},
$$

$$
\mathrm{Recall}_{\mathrm{macro}} = \frac{1}{C} \sum_{i=1}^{C} \frac{TP_i}{TP_i + FN_i},
$$

$$
\mathrm{F1\text{-}score}_{\mathrm{macro}} = \frac{1}{C} \sum_{i=1}^{C} \frac{2 \cdot TP_i}{2 \cdot TP_i + FP_i + FN_i},
$$

$$
\mathrm{Agreement} = \frac{\sum_{i=1}^{C} TP_i}{\sum_{i=1}^{C} (TP_i + FN_i)}.
$$

By convention, calibration is measured with respect to a set of discrete (binary) labels. To evaluate model calibration, we first binarize the preference distribution to obtain ground truth preference labels and then measure the Expected Calibration Error (ECE) and Brier Score. Test samples with ground truth label being "Tie" are skipped for calibration evaluation.

The ECE is calculated by dividing the confidence into $K$ equal-sized bins (each of size $\frac{1}{K}$), and then calculating the accuracy and average confidence within each bin:

$$\text{ECE} = \sum_{k=1}^{K} \frac{|B_k|}{N} \left| \text{Acc}(B_k) - \text{Conf}(B_k) \right|,$$

$$\text{Acc}(B_k) = \frac{1}{|B_k|} \sum_{i \in B_k} \mathbf{1}(\hat{y}_i = y_i), \qquad \text{Conf}(B_k) = \frac{1}{|B_k|} \sum_{i \in B_k} \hat{p}_i,$$

where $B_k$ is the number of samples whose prediction confidence falls into the interval $(\frac{k-1}{K}, \frac{k}{K}]$, $\hat{y}_i$ and $y_i$ are the predicted and true preference labels, and $\hat{p}_i$ is the predicted probability. By default, we set $K = 10$ in our experiments.

The Brier score is calculated as $\frac{1}{N} \sum_{i=1}^{N} (y_i - \hat{p}_i)^2$.

## C.2 FINETUNING DETAILS

In Setting 2, we optimize the following GRPO objective (Shao et al., 2024)

$$\mathcal{J}(\boldsymbol{\theta}) = \mathbb{E}_{\boldsymbol{x} \sim p(\boldsymbol{x}), \boldsymbol{\tau} \sim \boldsymbol{\pi}_{\boldsymbol{\theta}_{\text{old}}}} \left[ \min\left( \frac{\boldsymbol{\pi}_{\boldsymbol{\theta}}(\boldsymbol{\tau}|\boldsymbol{x})}{\boldsymbol{\pi}_{\boldsymbol{\theta}_{\text{old}}}(\boldsymbol{\tau}|\boldsymbol{x})} A_t, \; \text{clip}\left( \frac{\boldsymbol{\pi}_{\boldsymbol{\theta}}(\boldsymbol{\tau}|\boldsymbol{x})}{\boldsymbol{\pi}_{\boldsymbol{\theta}_{\text{old}}}(\boldsymbol{\tau}|\boldsymbol{x})}, \; 1-\varepsilon, \; 1+\varepsilon \right) A_t \right) - \beta \, D_{\text{KL}}(\boldsymbol{\pi}_{\boldsymbol{\theta}} \, \| \, \boldsymbol{\pi}_{\text{ref}}) \right],$$

with the unbiased KL estimator (Schulman, 2020)

$$D_{\text{KL}}(\boldsymbol{\pi}_{\boldsymbol{\theta}} \, \| \, \boldsymbol{\pi}_{\text{ref}}) \approx \frac{\boldsymbol{\pi}_{\text{ref}}(\boldsymbol{\tau}|\boldsymbol{x})}{\boldsymbol{\pi}_{\boldsymbol{\theta}}(\boldsymbol{\tau}|\boldsymbol{x})} - \log \frac{\boldsymbol{\pi}_{\text{ref}}(\boldsymbol{\tau}|\boldsymbol{x})}{\boldsymbol{\pi}_{\boldsymbol{\theta}}(\boldsymbol{\tau}|\boldsymbol{x})} - 1,$$

where $\boldsymbol{\pi}_{\boldsymbol{\theta}}$ is the policy model being optimized, $\boldsymbol{\pi}_{\boldsymbol{\theta}_{\text{old}}}$ is the previous policy model, $\boldsymbol{\pi}_{\text{ref}}$ is the reference policy, $A_t$ is the advantage estimate, $\varepsilon$ is the clipping hyperparameter, and $\beta$ is the KL penalty coefficient. Detailed parameter settings for our experiments are presented in Table 6.

# D ADDITIONAL RESULTS

## D.1 WIN RATE PREDICTION

We use a subset of the LMSys Chatbot Arena Conversation dataset (Zheng et al., 2023; Kahng et al., 2025) to evaluate the autorater's ability to predict the LM's win rate. This subset contains a total of 900 prompts for comparing responses from Gemma 1.0 and Gemma 1.1. For this evaluation, we use Qwen-2.5-7B as the base model for the autorater to avoid potential self-enhancement bias (i.e., favoring responses generated by LMs from the same family). Specifically, given two LMs $\boldsymbol{\pi}_A$ and $\boldsymbol{\pi}_B$, the win rate (of $\boldsymbol{\pi}_B$) is defined as

$$\Pr[\boldsymbol{\pi}_B \succ \boldsymbol{\pi}_A] := \mathbb{E}_{\boldsymbol{x} \sim p(\boldsymbol{x})} \mathbb{E}_{\boldsymbol{\tau}_A \sim \boldsymbol{\pi}_A, \boldsymbol{\tau}_B \sim \boldsymbol{\pi}_B} \Pr[\boldsymbol{\tau}_B \succ \boldsymbol{\tau}_A | \boldsymbol{x}].$$

From the results shown in Table 7, we observe that the predicted win rates from the finetuned autoraters are more aligned with the true win rate voted by human judges.

## D.2 RL WITH PROBABILISTIC LABELS

While our main experiments utilized sparse binary labels for RL, our reward function is also compatible with dense probabilistic labels. To explore how this data format affects performance, we finetuned Qwen-2.5-7B using the Brier score reward on the same 5K prompts (each with 10 annotations) used for SFT.

As shown in Figure 3, we compared this model against several baselines: the zero-shot model with direct verbalized confidence, the SFT model trained on probabilistic labels, and our primary model

**Table 6:** Detailed finetuning settings.

| Setting 1 — Supervised Finetuning (SFT) | |
|---|---|
| *General* | |
| max total sequence length | 2048 |
| precision | bf16 |
| *Optimization* | |
| optimizer | AdamW (Loshchilov & Hutter, 2019) |
| optimizer hyperparameters | $\beta_1, \beta_2, \epsilon = 0.9, 0.98, 10^{-5}$ |
| weight decay | 0.1 |
| batch size | 32 |
| training epochs | 3 |
| learning rate | $1 \times 10^{-6}$ |
| **Setting 2 — Reinforcement Learning (RL)** | |
| *General* | |
| max total sequence length | 2048 |
| precision | bf16 |
| *GRPO Setting* | |
| hyperparameters | $\beta = 0.01, \varepsilon = 0.2$ |
| number of prompts per step | 32 (Gemma) / 64 (Qwen) |
| number of generations per prompt | 32 |
| *Optimization* | |
| optimizer | AdamW (Loshchilov & Hutter, 2019) |
| optimizer hyperparameters | $\beta_1, \beta_2, \epsilon = 0.9, 0.999, 10^{-8}$ |
| weight decay | 0.001 |
| batch size | 512 |
| training epochs | 1 |
| learning rate | $3 \times 10^{-7}$ |
| learning rate warm-up | linear |
| warm-up ratio / steps | 0.01 / 50 |

**Table 7:** Evaluation on the Chatbot Arena Conversations dataset for comparing Gemma 1.0 to Gemma 1.1. We compare the autorater's predicted win rate for Gemma 1.0 against the true win rate from human annotators.

| Method | Win Rate of Gemma 1.0 | Absolute Error to Human |
|---|---|---|
| Human | 0.4344 | — |
| Qwen-2.5-7B Verbal | 0.7397 | 0.3053 |
| Qwen-2.5-7B Verbal w/ CoT | 0.5951 | 0.1607 |
| Qwen-2.5-7B SFT | 0.3146 | 0.1198 |
| Qwen-2.5-7B SFT w/ CoT | 0.3082 | 0.1262 |
| Qwen-2.5-7B RL (Brier) | 0.3640 | 0.0704 |
| Qwen-2.5-7B RL (Log) | **0.3662** | **0.0682** |

trained with RL on binary labels. The results indicate that RL with probabilistic labels improves both performance (agreement) and calibration (ECE) over the zero-shot baseline. However, it underperformed compared to the autoraters trained with a larger set of binary labels, highlighting the crucial role of data diversity in achieving optimal results.

### D.3 CONTROLLED BASELINE COMPARISONS

To isolate the benefits of our distribution-matching framework from the underlying data quality, we evaluated our method against two controlled baselines trained on the exact same data splits using the Qwen-2.5-7B model. The results are summarized in Table 8.

**Distributional SFT vs. Standard SFT.** Standard supervised finetuning (SFT) is typically trained on discrete hard labels. In our controlled baseline, we train the model on a single sampled annotation per prompt, simulating standard noise in data collection. We compared this against our

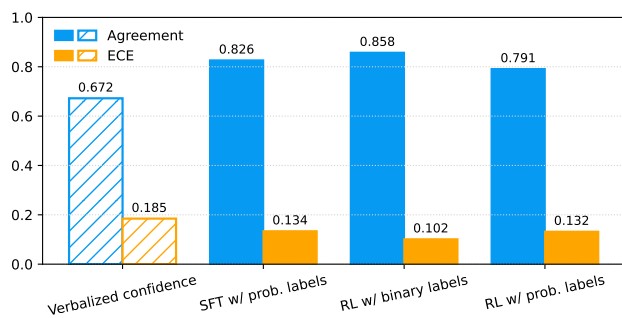

**Figure 3:** Result of RL finetuning with probabilistic labels.

distributional SFT, which aggregates available annotations into a soft target. While both models utilize the same underlying teacher signals, our distributional approach significantly improves alignment and calibration. Specifically, the alignment error (MSE) decreases from 0.1462 to 0.1143, and the calibration (Brier) score improves from 0.1766 to 0.1394, demonstrating that modeling the full probability spectrum reduces overconfidence even when derived from the same data source.

**Probabilistic RL vs. Bradley-Terry (BT) Reward Model.** Standard reward modeling typically assumes the Bradley-Terry (BT) model, where preference probabilities are derived from a scalar reward difference: $P[B \succ A | \boldsymbol{x}] = \sigma(r(\boldsymbol{x}, B) - r(\boldsymbol{x}, A))$. To compare with this BT baseline, we trained a reward model head using the same 50K prompts with binary preference labels used for our RL experiments. As shown in Table 8, our RL-Brier method, which models preferences as conditionally independent Bernoulli trials without enforcing the transitivity constraints of BT, achieves a significantly better fit to the population distribution. Our method reduces the Brier score by over 50% compared to the BT baseline ($0.2336 \rightarrow 0.1103$) and drastically improves agreement ($59.67\% \rightarrow 85.75\%$), confirming that this flexible formulation captures complex disagreement patterns more effectively.

**Table 8:** Controlled baseline comparison on Qwen-2.5-7B across two data regimes.

| Method | Objective Type | Alignment (MSE) ↓ | Performance (Agr.) ↑ | Calibration (Brier) ↓ |
|---|---|---|---|---|
| *Data Regime 1: Multi-Annotator Source* | | | | |
| Standard SFT | SFT on Single Sample | 0.1462 | 0.7993 | 0.1766 |
| Dist. SFT (Ours) | SFT on Aggregated Dist. | 0.1143 | 0.8264 | 0.1394 |
| *Data Regime 2: Single-Annotator Source* | | | | |
| Bradley-Terry Model | Supervised Ranking Loss | 0.1813 | 0.5967 | 0.2336 |
| RL-Brier (Ours) | RL with Proper Scoring Rule | 0.0893 | 0.8575 | 0.1103 |

## D.4 Output Parsability Analysis

A potential concern with reinforcement learning on open-ended generation is the degradation of output structure (e.g., losing the required XML tags). To verify the structural robustness of our trained autoraters, we measured the empirical parsability rate, defined as the percentage of test-time generations that can be successfully parsed into a valid numeric probability $p \in [0, 1]$ by our deterministic parser.

Across all experiments on JudgeLM, our RL-finetuned models (both Brier and Log rewards) maintained a 100% parsability rate. Our SFT models similarly achieved > 99.8% parsability. This confirms that the penalty term for unparsable outputs in our reward function effectively constrains the policy to the valid output format while optimizing for calibration.

## E Discussions

We provide a discussion on the extensions and limitations of our approach and results.

First, our work initiates the study on more reliable autoraters by predicting the full preference distribution. While our empirical results focused on the pairwise evaluation setting, the general framework and analysis also apply to the pointwise evaluation setting, where the preference takes nominal or ordinal values (e.g., Likert scale). Specifically, a direction extension involves employing multiclass strictly proper scoring rules, such as the multi-class Brier score or cross-entropy, as the reward function for RL. We leave a thorough empirical study in this space as future work.

Second, our finetuning objective aims to capture the distribution of human preferences, which stems from the human-level uncertainty on the subject. As such, the uncertainty communicated by the finetuned autorater in the form of verbalized confidence is rather aleatoric than epistemic, whereas the latter may require predicting a second-order distribution (i.e., a distribution over the predicted probability $p_{\boldsymbol{\theta}}(\boldsymbol{x})$). However, such an approach may significantly complicate training; instead, we demonstrate empirically that simply providing a point estimation of the probability is sufficient to improve the autorater's alignment to the preference distribution while maintaining good out-of-distribution generalizability.

## F  LLM USAGE

LLMs have been used to generate surrogate preference labels in our experiments and to assist and improve the writing of this paper.

## G  REPRODUCIBILITY STATEMENT

To ensure our work is reproducible, we provide a detailed account of our experimental setup in Section 4. Full implementation details, including finetuning parameters and evaluation procedures, are available in Appendix C. All prompts used for annotation and evaluation are included in Appendix I. We intend to make our source code and data publicly available upon acceptance of the manuscript.

## H  ETHICS STATEMENT

Our research contributes to AI alignment by proposing a shift from training autoraters on discrete preference labels to predicting the full distribution of human preferences. This method allows the autoraters to capture a wider spectrum of human opinion, which can lead to fairer and more reliable AI systems that better serve societal welfare.

While the goal is to create more aligned AI, we recognize that any model can be misused or inherit biases present in the annotation data. The preference distributions captured by our model reflect the demographics of the annotators. Therefore, care must be taken to ensure this group is representative to avoid encoding societal biases. We believe that transparently modeling preference distributions is a step toward identifying and mitigating such issues.

## I  PROMPTS

---

**Prompt Set for Persona-Based Preference Annotation**

**Base Prompt Structure:**

```
{Persona-specific instruction here}
Your output must be a single letter, either 'A' or 'B', enclosed in <winner>
and </winner> tags.
[Question] {question}
[The Start of Response A] {responseA} [The End of Response A]
[The Start of Response B] {responseB} [The End of Response B]
Considering these factors, which response is better?
```

---

**Persona-Specific Instructions:**

**1. Factual & Technical Specialist** Your task is to evaluate two responses based on their helpfulness, relevance, accuracy, and level of detail. As a technical specialist, you would additionally consider **factual correctness and precision** in your judgment.

**2. Safety & Privacy Advocate** Your task is to evaluate two responses based on their helpfulness, relevance, accuracy, and level of detail. As a safety advocate, you would additionally consider **safety and ethical responsibility**. An unsafe response is a poor response.

**3. Empathetic Communicator** Judge the overall quality of the responses based on their helpfulness, accuracy, and detail. As an empathetic communicator, you would also pay close attention to the **tone and respectfulness** of the language.

**4. Concision Champion** Assess which response is better, considering its helpfulness, accuracy, and relevance. As a champion of concision, you would additionally prefer the response that is more **direct and to-the-point**, penalizing unnecessary verbosity.

**5. Nuance & Depth Seeker** Evaluate the helpfulness, relevance, and accuracy of both responses. As an academic researcher, you would also give importance to the **depth and nuance** of the explanation.

**6. Creative & Arts Critic** Assess which response is more helpful, relevant, and accurate. As a creative critic, you would also weigh the **originality, style, and engagement** of the writing.

**7. Everyday User (4x Weight)** From the perspective of a regular person, not an expert, evaluate which response is better overall. Your decision should be based on which one is most **helpful, relevant to the question, accurate, and provides a good level of detail**. The best response is the one that is most satisfying and clear.

---

**Prompts for Eliciting Probabilistic Prediction**

1. VERBALIZED CONFIDENCE

```
Your task is to evaluate two responses, A and B, based on the provided question.
    Compare the responses and determine the probability that **Response B is better
    than Response A**.

Your output must be a single float between 0.0 and 1.0, enclosed in `<prob_rb_better>`
    and `</prob_rb_better>` tags. Do not output any other text, reasoning, or
    explanations.
- A probability of 0.0 means Response A is clearly better.
- A probability of 1.0 means Response B is clearly better.
- A probability of 0.5 indicates they are of similar quality.

Example Output Format:
<prob_rb_better>0.85</prob_rb_better>

[Question]
{question}

[The Start of Response A]
{responseA}
[The End of Response A]

[The Start of Response B]
{responseB}
[The End of Response B]

Probability Output:
```

---

2. VERBALIZED CONFIDENCE W/ COT

```
Your task is to evaluate two responses, A and B, based on the provided question.
    Compare the responses on helpfulness, accuracy, and clarity.

Follow these steps for your output:
1. First, provide a step-by-step reasoning of your evaluation inside `<think>` and `</
    think>` tags. Explain which response is better and why.
```

```
2. Immediately after the closing '</think>' tag, provide a single float between 0.0 and
      1.0 representing the probability that **Response B is better than Response A**.
      This float must be enclosed in '<prob_rb_better>' and '</prob_rb_better>' tags. A
      probability of 0.0 means Response A is clearly better, 1.0 means Response B is
      clearly better, and 0.5 indicates they are of similar quality.

Example Output Format:
<think>Your detailed reasoning comparing A and B goes here.</think><prob_rb_better
      >0.85</prob_rb_better>

[Question]
{question}

[The Start of Response A]
{responseA}
[The End of Response A]

[The Start of Response B]
{responseB}
[The End of Response B]

Your Evaluation:
```

## 3. SELF-CONSISTENCY

```
Your task is to evaluate two responses, A and B, based on the provided question. You
      must decide which response is superior.
First, provide a step-by-step thinking process that justifies your decision. This
      reasoning should be enclosed within <think></think> tags.
After your reasoning, provide your final decision, which must be a single character ('A
      ' or 'B'), enclosed within <answer></answer> tags.

Example Output Format:
<think>Response A directly answers the user's question about X, while Response B is
      more general. Response A also provides a concrete example which is helpful.
      Therefore, A is the better response.</think><answer>A</answer>

[Question]
{question}

[The Start of Response A]
{responseA}
[The End of Response A]

[The Start of Response B]
{responseB}
[The End of Response B]

Final Evaluation:
```

## 4. LOGIT-BASED CONFIDENCE

```
Your task is to evaluate two responses, A and B, based on the provided question.
You must decide which response is superior. Your output must be a single character: 'A'
      or 'B'.
Do not output any other text, reasoning, or explanations.

Example Output Format:
B

[Question]
{question}

[The Start of Response A]
{responseA}
[The End of Response A]

[The Start of Response B]
{responseB}
[The End of Response B]

Final Decision (A or B):
```

# J  RESPONSE EXAMPLES

## J.1  JUDGELM DATASET

---

**Gemma-2-9B (JudgeLM *val* set - QID 827)**

- **Question**

Compose a haiku with the provided subject. Spring

- **Model A (vicuna-13b-1.1) Response**

```
Fresh new start, green leaves grow
Raindrops pitter-patter, flowers bloom.
```

- **Model B (mix-instruct-output) Response**

```
Amidst blooming flowers
Gentle breeze moves budding trees
Spring awakens life.
```

- **Ground Truth**

$P(\text{B is better}) = 1.00$

- **Autoraters Outputs**

**Verbalized Confidence w/ CoT (Prediction: 0.60)**

```
<think>Both responses follow the 5-7-5 syllable structure of a
    haiku.

Response B is slightly better because it paints a more vivid
    picture with its imagery. "Amidst blooming flowers" and
    "Gentle breeze moves budding trees" create a stronger sense
    of the season's atmosphere compared to Response A's more
    general descriptions.  Response B also uses more evocative
    language like "awakens life," which adds a deeper layer of
    meaning. </think><prob_rb_better>0.6</prob_rb_better>
```

**SFT w/ CoT (Prediction: 1.00)**

```
<think>1.  **Analyze Response A:** Response A attempts to follow
    the 5-7-5 syllable structure but fails. "Fresh new start,
    green leaves grow" has 7 syllables, "Raindrops
    pitter-patter, flowers bloom" has 9 syllables. It also
    includes a line break within the second line, which is not a
    haiku characteristic.
2.  **Analyze Response B:** Response B correctly follows the
    5-7-5 syllable structure for all three lines. It also uses
    evocative imagery and language appropriate for the subject
    "Spring."
3.  **Compare and Justify Score:** Response B is a perfect
    haiku, adhering to the strict structural and thematic
    requirements. Response A completely fails on the syllable
    count and line structure, making it not a haiku at all.
    Therefore, Response B is unequivocally better than Response
    A. A confidence score of 1.00 is warranted as there is no
    scenario where Response A would be preferred for this
    task.</think><prob_rb_better>1.0</prob_rb_better>
```

**RL (Brier) (Prediction: 0.99)**

```
<think>Response B is a better haiku because it follows the
    traditional structure of a haiku (5-7-5 syllables per line)
    while Response A does not.  Response B also captures the
    essence of spring with its imagery of blooming flowers,
    gentle breezes, and awakening life. Response A's lines are
    too long and don't have the same poetic flow.
    </think><prob_rb_better>0.99</prob_rb_better>
```

**RL (Log) (Prediction: 1.00)**

```
<think>Response B is better because it adheres to the 5-7-5
    syllable structure of a haiku, while Response A does not.
    Response B also uses more evocative language and imagery to
    capture the essence of spring. Response A is more of a
    descriptive paragraph than a
    haiku.</think><prob_rb_better>1.0</prob_rb_better>
```

**Gemma-2-9B (JudgeLM *val* set - QID 3960)**

- **Question**

Tell me what are aspects of polytopes from given text In geometry, a simplicial polytope is a polytope whose facets are all simplices. For example, a simplicial polyhedron in three dimensions contains only triangular faces and corresponds via Steinitz's theorem to a maximal planar graph.

They are topologically dual to simple polytopes. Polytopes which are both simple and simplicial are either simplices or two-dimensional polygons.

- **Model A (mix-instruct-output) Response**

```
1. A simplicial polytope is a polytope in geometry where each
    facet is a simplice.
2. As an illustration, a simplicial polyhedron in three
    dimensions with only triangular faces corresponds to a
    maximum planar network according to Steinitz's theorem.
3. They are simple polytopes topological dual.
4. Two-dimensional polygons or simplices are polytopes that are
    both simple and simplicial.
```

- **Model B (alpaca-native) Response**

```
The aspects of polytopes from given text are: a polytope whose
    facets are all simplices and topologically dual to simple
    polytopes. Polytopes which are both simple and simplicial
    are either simplices or two-dimensional polygons.
```

- **Ground Truth**

$P(\text{B is better}) = 0.90$

- **Autoraters Outputs**

**Verbalized Confidence w/ CoT (Prediction: 0.70)**

```
<think>Both responses attempt to summarize the aspects of
    polytopes from the provided text.

Response A breaks down the information into numbered points,
    making it slightly easier to read and understand. However,
    it contains some inaccuracies:

* Point 3 is incomplete and grammatically incorrect. It should
    read "They are topologically dual to simple polytopes."
* Point 2 uses the phrase "maximum planar network" which is
    likely a typo and should be "maximal planar graph."

Response B, while less structured, accurately summarizes the key
    aspects of polytopes mentioned in the text. It is concise
    and avoids the errors present in Response A.
    </think><prob_rb_better>0.7</prob_rb_better>
```

**SFT w/ CoT (Prediction: 0.40)**

<think>1.  **Analyze the Question:** The user wants to extract "aspects of polytopes" from the provided text. This implies identifying key definitions, properties, or relationships related to polytopes.

2.  **Evaluate Response A:**
    *   **Helpfulness & Relevance:** Response A attempts to list aspects. It correctly identifies "simplicial polytope" and "triangular faces." However, it includes "maximum planar network" instead of "maximal planar graph," which is a minor factual error. It also lists "topological dual" and "two-dimensional polygons or simplices."
    *   **Coherence & Quality:** Response A is presented as a numbered list, which is a good format for extracting distinct points. The language is clear and directly quotes or paraphrases the text.

3.  **Evaluate Response B:**
    *   **Helpfulness & Relevance:** Response B attempts to list aspects. It correctly identifies "a polytope whose facets are all simplices" and "topologically dual to simple polytopes." It also includes "Polytopes which are both simple and simplicial are either simplices or two-dimensional polygons."
    *   **Coherence & Quality:** Response B is a single, concise sentence. While it successfully condenses the information, it lacks the explicit separation of distinct aspects that a numbered list provides. It also uses the phrase "The aspects of polytopes from given text are:" which is a bit redundant given the question.

4.  **Compare and Justify Score (0.40):**
    *   Response A is slightly better because it presents the information in a more structured and easily digestible format (a numbered list). This makes it clearer that each point is a distinct "aspect."
    *   Response B is more concise but less organized. Its single sentence format makes it harder to parse as a list of distinct aspects.
    *   The minor factual error in Response A ("maximum planar network" instead of "maximal planar graph") is not significant enough to outweigh the superior presentation and clarity of Response A.
    *   Therefore, Response A is marginally better, leading to a confidence score of 0.40 (meaning B is better with 40% *probability, or A is better with 60%* *probability).*</think><prob_rb_better>0.4</prob_rb_better>

**RL (Brier) (Prediction: 0.98)**

<think>Response B is better because it directly answers the question in a concise and clear manner. It identifies the key aspects of polytopes mentioned in the text without unnecessary elaboration or rephrasing. Response A, while accurate, provides a more verbose and less focused summary of the text. </think><prob_rb_better>0.98</prob_rb_better>

**RL (Log) (Prediction: 0.65)**

```
<think>Response B is better because it directly answers the
    question by extracting and summarizing the key aspects of
    polytopes mentioned in the text. Response A, while accurate,
    simply rephrases the information from the text without
    providing a clear and concise summary. Response B is more
    focused and highlights the essential concepts related to
    polytopes. </think><prob_rb_better>0.65</prob_rb_better>
```

**Qwen-2.5-7B (JudgeLM *val* set - QID 376)**

● **Question**

What is an anemone?

● **Model A (alpaca-native) Response**

```
An anemone is a type of sea creature that is typically found in
    marine habitats. It is typically described as a large,
    beautiful, and often colorful animal that is found on coral
    reefs. Anemones are typically classified as either solitary
    or colonial, depending on the species. Solitary anemones are
    typically smaller and are found in shallow waters, while
    colonial anemones are larger and can be found in deeper
    waters. The anemone uses its stinging cells to capture prey,
    and is also known to benefit from the symbiotic relationship
    it has with certain types of algae.
```

● **Model B (vicuna-13b-1.1) Response**

```
An anemone is a type of marine animal that belongs to the phylum
    Cnidaria and the class Anthozoa. It is a multicellular,
    radially symmetrical animal that has a simple, tubular body
    and a single, central oral-aboral opening surrounded by
    tentacles. Anemones are found in coral reefs, on the
    seafloor, and in shallow waters. They are predators, feeding
    on a variety of small sea creatures. An anemone's color can
    range from brown to red, orange, or purple.
```

● **Ground Truth**

$P(\text{B is better}) = 0.70$

● **Autoraters Outputs**

**Verbalized Confidence w/ CoT (Prediction: 0.95)**

```
<think>
Response A provides a basic description of an anemone,
    mentioning its habitat and classification into solitary or
    colonial types. However, it lacks detail about the anemone's
    physical characteristics and behavior. Response B, on the
    other hand, gives a more detailed and scientifically
    accurate description, mentioning the phylum and class of
    anemones, their physical structure, and feeding habits. It
    also provides a range of colors, which is more specific and
    helpful for understanding the anemone's appearance.
    Therefore, Response B is more accurate and detailed.
</think>
<prob_rb_better>0.95</prob_rb_better>
```

**SFT w/ CoT (Prediction: 0.90)**

```
<think>1.  **Analyze Response A:** Response A describes an
    anemone as a "large, beautiful, and often colorful animal."
    This is a significant factual error. Anemones are typically
    small, often less than a few inches in diameter, and not
    generally described as "large." This misrepresentation of
    size is a major flaw.
2.  **Analyze Response B:** Response B provides a more accurate
    and detailed description. It correctly identifies anemones
    as belonging to the phylum Cnidaria and class Anthozoa, and
    describes their physical characteristics (multicellular,
    radially symmetrical, tubular body, tentacles, oral-aboral
    opening). It also correctly states their habitat and feeding
    habits.
3.  **Compare Helpfulness & Relevance:** Response B is far more
    helpful and relevant due to its accuracy. Response A's
    fundamental error regarding size makes it misleading and
    unhelpful.
4.  **Compare Coherence & Quality:** Response B is
    well-structured, clear, and accurate. Response A contains a
    critical factual error that undermines its quality.
5.  **Justify Score (0.90):** Response B is significantly better
    because it provides accurate and comprehensive information,
    while Response A contains a major factual error regarding
    the size of an anemone. The error in Response A makes it
    largely unhelpful and potentially misleading. Therefore,
    Response B is overwhelmingly
    superior.</think><prob_rb_better>0.9</prob_rb_better>
```

**RL (Brier) (Prediction: 0.99)**

```
<think>
Response A is less accurate and less clear. It incorrectly
    describes anemones as "beautiful" and "large," which are not
    typical descriptors for anemones. It also incorrectly states
    that solitary anemones are typically smaller and found in
    shallow waters, while colonial anemones are larger and found
    in deeper waters, which is not a general rule. Response B,
    on the other hand, provides a more accurate and clear
    description, mentioning that anemones belong to the phylum
    Cnidaria and class Anthozoa, and correctly describes their
    physical characteristics and habitat. It also correctly
    states that anemones are predators and that their color can
    vary.

Given these differences, Response B is clearly better.
</think><prob_rb_better>0.99</prob_rb_better>
```

**RL (Log) (Prediction: 1.00)**

```
<think>
Response A is less accurate and less clear. It incorrectly
    describes anemones as "beautiful" and "large," which is not
    a defining characteristic. It also incorrectly states that
    solitary anemones are typically smaller and found in shallow
    waters, while colonial anemones are larger and found in
    deeper waters, which is not a general rule. Response B, on
    the other hand, provides a more precise and accurate
    description, including the scientific classification and the
    correct habitat and feeding habits. It also correctly
    mentions the range of colors anemones can have.

</think><prob_rb_better>1.0</prob_rb_better>
```

---

### Qwen-2.5-7B (JudgeLM *val* set - QID 3311)

- **Question**

How can I get rid of a rash on my neck?

- **Model A (moss-moon-003-sft) Response**

```
There are several ways to get rid of a rash on your neck. Here
    are some options:

1. Try over-the-counter creams: Over-the-counter creams like
    hydrocortisone cream or chapstick can help soothe and
    moisturize the rash. Be sure to read the labels carefully
    and follow the instructions.

2. Use aloe vera: Aloe vera juice can help soothe and moisturize
    the skin, making it less itchy. Mix one tablespoon of aloe
    vera gel with one tablespoon of water and apply to the rash.

3. Use tea tree oil: Tea tree oil has anti-inflammatory
    properties that can help reduce inflammation and itching.
    Apply a few drops of tea tree oil to the rash and leave it
    on for 30 minutes before rinsing it off.

4. Get a rash shot: In some cases, your doctor may recommend a
    rash shot to help reduce the itching and inflammation. This
    is a prescription medication that you should consult your
    doctor about before taking.

5. Try home remedies: There are several home remedies that can
    help soothe a rash on your neck. For example, you can try
    applying cucumber slices to the rash to help
```

- **Model B (oasst-sft-4-pythia-12b-epoch-3.5) Response**

```
You should see an allergist to get tested for allergies.
```

- **Ground Truth**

$P(\text{B is better}) = 0.50$

- **Autoraters Outputs**

**Verbalized Confidence w/ CoT (Prediction: 0.15)**

```
<think>
Response A provides a detailed list of methods to alleviate a
    rash on the neck, including over-the-counter creams, aloe
    vera, tea tree oil, rash shots, and home remedies. It offers
    a comprehensive approach and is helpful and clear for
    someone dealing with a rash. Response B, on the other hand,
    suggests seeing an allergist, which is a more specific and
    targeted approach but does not provide immediate relief or
    practical steps for managing the rash. While Response B is
    accurate in suggesting a professional consultation, it lacks
    the practical, immediate advice that Response A provides.

Given the context of the question, which asks for ways to get
    rid of a rash, Response A is more helpful and comprehensive.
    However, Response B is still accurate and could be a
    valuable follow-up step after trying the methods in Response
    A.
</think>
<prob_rb_better>0.15</prob_rb_better>
```

**SFT w/ CoT (Prediction: 0.90)**

```
<think>1.  **Analyze Response A:** Response A provides a list of
    potential solutions, including OTC creams, aloe vera, tea
    tree oil, and even a "rash shot." While some of these
    suggestions (like tea tree oil) are generally safe for
    topical use, others (like "rash shot") are vague and
    potentially misleading without proper medical consultation.
    The response also cuts off abruptly.
2.  **Analyze Response B:** Response B suggests seeing an
    allergist. This is a highly relevant and potentially crucial
    step for diagnosing and treating a rash, especially if it's
    allergy-related. It prioritizes professional medical advice.
3.  **Compare Helpfulness & Relevance:** Response B is
    significantly more helpful and relevant because it points
    the user towards a professional diagnosis, which is often
    the first and most important step in treating a rash.
    Response A's suggestions, while not inherently harmful, are
    less direct and potentially less effective without a proper
    diagnosis.
4.  **Compare Coherence & Quality:** Response A is a bit
    disjointed due to the abrupt ending and the inclusion of
    potentially confusing or vague advice (e.g., "rash shot").
    Response B is concise and directly addresses the underlying
    cause, making it more coherent and actionable.
5.  **Justify Score (0.90):** Response B is overwhelmingly
    better because it prioritizes professional medical advice,
    which is the most responsible and effective way to address a
    rash. Response A's suggestions, while not actively harmful,
    are less direct and potentially less effective without a
    proper diagnosis. The score of 0.90 reflects that B is
    significantly superior due to its focus on professional
    guidance.</think><prob_rb_better>0.9</prob_rb_better>
```

**RL (Brier) (Prediction: 0.75)**

```
    <think>
    Response A provides a detailed list of methods to treat a rash
        on the neck, including over-the-counter creams, aloe vera,
        tea tree oil, and home remedies. However, it is quite
        lengthy and includes some less conventional methods like
        "rash shots" which are not commonly used for rashes and
        could be misleading. The response is clear but could be more
        concise.

    Response B, on the other hand, suggests seeing an allergist,
        which is a more direct and professional approach. This is
        particularly useful if the rash is due to an allergic
        reaction. However, it doesn't provide any specific treatment
        options and is quite brief.

    Given the question, Response A offers more comprehensive
        information, but Response B provides a more targeted and
        professional approach. Since the question asks for a
        specific recommendation, Response B is more likely to be
        helpful for the user.

    Considering the above, Response B is better, but not by a wide
        margin.
    </think><prob_rb_better>0.75</prob_rb_better>
```

**RL (Log) (Prediction: 0.00)**

```
    <think>
    Response A provides a detailed list of methods to treat a rash
        on the neck, including over-the-counter creams, aloe vera,
        tea tree oil, and home remedies. It is clear and offers
        multiple options, making it helpful for the user. Response
        B, on the other hand, suggests seeing an allergist, which is
        a single action and does not provide specific treatment
        options. While it might be a valid step in some cases, it
        does not directly address the user's question about how to
        get rid of the rash.

    Given the comprehensive nature and directness of Response A, it
        is more helpful and clearer for the user's needs.
    </think>
    <prob_rb_better>0.0</prob_rb_better>
```

## J.2 PANDALM DATASET

**Gemma-2-9B** (PandaLM *test* set - QID 412)

- **Question**

Instruction: Rewrite the given text and correct grammar, spelling, and punctuation errors.
Input: If you'd told me year ago that today I would finish a marathon, I would of laughed. Your support had a huge affect on me!

- **Model A (bloom-7b) Response**

```
    If you told me a year ago that today I would finish a marathon,
        I would have laughed. Your support had a huge effect on me!
```

- **Model B (cerebras-gpt-6.7B) Response**

> If you had told me a year ago that today I would finish a
>    marathon, I would have laughed. Your support had a huge
>    impact on me.

- **Ground Truth**

Preference: B better

- **Autoraters Outputs**

**Verbalized Confidence w/ CoT (Prediction: 0.60)**

```
<think>Response B is slightly better because it correctly uses
    the past perfect tense ("had told") which is more
    grammatically accurate in this context. Response A uses the
    simple past tense ("told") which is less precise.
    Additionally, Response B uses the word "impact" which is a
    more precise synonym for
    "affect".</think><prob_rb_better>0.6</prob_rb_better>
```

**SFT w/ CoT (Prediction: 0.60)**

```
<think>1. **Identify Errors in Input:** The input contains two
    main errors: "would of" should be "would have," and "affect"
    should be "effect" or "impact."
2. **Evaluate Response A:** Response A correctly changes "would
    of" to "would have" and "affect" to "effect."
3. **Evaluate Response B:** Response B also correctly changes
    "would of" to "would have" and "affect" to "impact."
4. **Compare Responses:** Both responses correctly identify and
    fix the grammatical errors. However, Response B uses
    "impact" instead of "effect," which is a more common and
    widely accepted synonym for "affect" in this context
    (meaning influence). While "effect" is also grammatically
    correct, "impact" is arguably a slightly more precise and
    natural fit for the meaning of "huge affect."
5. **Justify Score:** Response B is marginally better due to
    its choice of "impact" over "effect," making the corrected
    sentence slightly more precise and natural. The difference
    is subtle, but it gives B a slight edge in quality.
    Therefore, a confidence score of 0.60 is appropriate,
    indicating B is slightly
    better.</think><prob_rb_better>0.6</prob_rb_better>
```

**RL (Brier) (Prediction: 0.98)**

```
<think>Response B is better because it correctly uses the past
    perfect tense ("had told") and the word "impact" instead of
    "affect," which is used incorrectly in the original text.
    Response A is grammatically correct but uses the incorrect
    past perfect tense ("told") and the word "effect" which is
    not the best choice in this context.
    </think><prob_rb_better>0.98</prob_rb_better>
```

**RL (Log) (Prediction: 1.00)**

```
<think>Response B is better because it makes more accurate
    grammatical corrections.

*   Response A correctly changes "would of" to "would have" but
    keeps "affect" which should be "effect".
* Response B correctly changes "would of" to "would have" and
    "affect" to "impact".

Response B is also slightly more natural in its phrasing.
    </think><prob_rb_better>0.999</prob_rb_better>
```

## Qwen-2.5-7B (PandaLM *test* set - QID 573)

● **Question**

Instruction: Predict how many stars the author will give to the restaurant from a Yelp review.
Input: The reviews were great, but honestly i felt it was just ok. Seemed like a typical tourist
spot were the food was mediocre. The service was ok, not great, but not the worst. The bread
was very good. I ordered the muscle fries. They had a sweet Vidalia onion flavor. No need for
fries bc they were soggy wo much flavor. My boyfriend ordered a pesto chicken pasta and it was
again mediocre.

● **Model A (cerebras-gpt-6.7B) Response**

```
5 stars
```

● **Model B (opt-7b) Response**

```
The author will likely give the restaurant 3 stars.
```

● **Ground Truth**

Preference: B better

● **Autoraters Outputs**

**Verbalized Confidence w/ CoT (Prediction: 0.95)**

```
<think>
Response A simply states "5 stars" without any reasoning or
    context, which is not helpful or accurate given the mixed
    feedback in the review. Response B, on the other hand,
    provides a more reasoned prediction based on the content of
    the review. The review mentions that the experience was
    "just ok" and that the food and service were "mediocre,"
    which suggests a below-average experience. Therefore, a
    3-star rating seems more appropriate than 5 stars.

Response B is clearer and more aligned with the sentiment
    expressed in the review.
</think>
<prob_rb_better>0.95</prob_rb_better>
```

**SFT w/ CoT (Prediction: 1.00)**

```
<think>1.  **Analyze the Input:** The review is mixed. It starts
    with "The reviews were great, but honestly i felt it was
    just ok." This indicates a neutral to slightly negative
    overall impression. It then lists specific criticisms
    (mediocre food, mediocre service, soggy fries) and positive
    points (good bread, sweet Vidalia onion flavor). The
    language is generally unenthusiastic.
2.  **Evaluate Response A:** Response A predicts "5 stars." This
    is highly unlikely given the content of the review. The
    reviewer explicitly states it was "just ok" and lists
    multiple mediocre aspects. A 5-star rating would imply
    strong satisfaction, which is not evident.
3.  **Evaluate Response B:** Response B predicts "3 stars." This
    is a much more plausible rating for the given review. A
    3-star rating typically indicates a neutral to slightly
    negative experience, which aligns well with the mixed and
    somewhat critical tone of the review.
4.  **Compare Responses:** Response B's prediction is
    significantly more accurate and reasonable than Response
    A's. Response A is completely off, while Response B is a
    good, educated guess based on the review's content.
5.  **Justify Score:** A confidence score of 1.00 is appropriate
    because Response B is definitely better than Response A.
    Response A's prediction is entirely incorrect and unhelpful,
    whereas Response B provides a plausible and well-reasoned
    prediction.</think><prob_rb_better>1.0</prob_rb_better>
```

**RL (Brier) (Prediction: 0.99)**

```
<think>
Response A is overly optimistic and does not reflect the mixed
    sentiment in the review. The review mentions "great reviews"
    but also states the experience was "just ok" and "mediocre."
    The bread was good, but the fries and pasta were described
    as "soggy" and "mediocre." Given the overall mixed and
    somewhat negative tone, a 5-star rating is not appropriate.

Response B, on the other hand, suggests a 3-star rating, which
    seems more aligned with the mixed sentiment expressed in the
    review. A 3-star rating typically indicates an average
    experience, which fits the description of the restaurant
    being "just ok" and having "mediocre" food and service.

Therefore, Response B is better than Response A.
</think><prob_rb_better>0.99</prob_rb_better>
```

**RL (Log) (Prediction: 1.00)**

```
<think>
Response A is overly optimistic and does not reflect the overall
    sentiment of the review. The review mentions that the food
    was "mediocre" and the service was "ok, not great." There is
    no indication that the author would give 5 stars. Response
    B, on the other hand, is more aligned with the sentiment
    expressed in the review. The author found the experience to
    be "just ok" and the food "mediocre," which suggests a
    middle rating. Therefore, Response B is more accurate and
    helpful.
</think>
<prob_rb_better>1.0</prob_rb_better>
```

