# OpenReview forum: "Judging with Confidence: Calibrating Autoraters to Preference Distributions"
_ICLR.cc/2026/Conference — Submitted to ICLR 2026_

### Official Review · Reviewer_ZXxy · 2025-10-22

**Soundness:** 2
**Presentation:** 3
**Contribution:** 2
**Rating:** 2
**Confidence:** 3

**Summary:**

This paper argues that automated LLM judges ("autoraters") should be trained to output a probability distribution over preferences, rather than a single discrete label. This is motivated by the fact that many evaluation tasks are subjective, ambiguous, or nuanced, and collective human judgment is better represented by a distribution than a single "ground truth".

The authors formalize the target as a population-level Bernoulli parameter $p^*(x) = \Pr(B \succ A \mid x)$ for a given pair of responses $(A, B)$. They propose two general methods to finetune an autorater to verbalize this probability:
1.  **Supervised Finetuning (SFT):** Used when dense, probabilistic labels (e.g., from multiple annotators) are available.
2.  **Reinforcement Learning (RL):** Used when only sparse, binary labels are available. This approach uses proper scoring rules (specifically, Brier or Log rewards) to optimize the policy.

The authors create a synthetic "ground truth" distribution by prompting Gemini-2.5-Flash with various personas. Empirically, their finetuned models (especially RL with a Brier reward) show improved alignment (lower MSE) and calibration (lower ECE) against this target distribution, a significant reduction in positional bias, and strong, competitive performance on objective evaluation benchmarks like JudgeBench.

**Strengths:**

1.  **Clear and Important Problem Framing:** The paper provides a clear motivation for moving beyond discrete labels in LLM evaluation. It correctly identifies that collapsing subjective, population-level disagreement into a single "majority vote" label discards crucial information about uncertainty and minority viewpoints. The proposal to model the full preference distribution is a well-motivated and important direction.
2.  **Principled Training Methods:** The paper proposes two distinct and practical training strategies tailored to different, realistic data availability scenarios (dense probabilistic vs. sparse binary labels) . The use of proper scoring rules (Brier, Log) for the RL objective is theoretically sound, and the authors provide a proof of Fisher consistency (Proposition 1).
3.  **Strong Results on Positional Bias:** A key strength is the rigorous analysis of positional bias. The paper uses clear metrics (consistency and symmetry deviation)  and demonstrates that the distribution-matching objective, particularly with the RL-Brier reward, nearly eliminates this bias, achieving near-perfect swap-symmetry (Table 2).
4.  **Data Efficiency Insight:** The finding that RL with many sparse binary labels is more data-efficient than SFT with fewer dense probabilistic labels (for a fixed annotation budget) is a valuable practical insight, highlighting the importance of prompt diversity.

**Weaknesses:**

1.  **Arbitrary "Ground Truth" Distribution:** The entire framework's success is benchmarked against a "ground truth" preference distribution $p^*(x)$. However, this distribution is synthetically constructed using a single teacher model (Gemini-2.5-Flash) prompted with a specific, ad-hoc set of 7 personas and hard-coded weights (e.g., "Everyday User (4x Weight)" ) at a high temperature ($T=1.0$). This target distribution seems highly design-dependent and arbitrary. It is unclear whether this target represents a meaningful, general-purpose population preference or merely a specific artifact of this particular prompt-engineering process.
2.  **Ignores Preference Structure and Transitivity:** The paper models each response pair $(A, B)$ as an independent Bernoulli trial. While the authors present this as an advantage that avoids the Bradley-Terry (BT) assumption, it also discards a massive amount of structural information. This formulation does not enforce or model transitivity (i.e., if $A \succ B$ and $B \succ C$, then $A \succ C$). This structure is fundamental to preference learning and is exploited by standard reward models to improve sample efficiency and logical consistency. *The paper provides no analysis of its model's transitivity violations nor any comparison against standard BT-style reward model baselines.*
3.  **Training and Evaluation Data Overlap:** The main results for alignment (MSE) and calibration (ECE) in Table 1 are reported on a test set where the labels were generated by the *exact same* Gemini-2.5-Flash + persona methodology used to create the training data. While OOD results are provided later, these headline results may primarily demonstrate successful *imitation* of the teacher's specific (and arbitrary) label distribution, rather than a more general improvement in judgment or calibration.
4.  **No Downstream Task Validation:** The paper strongly motivates the need for probabilistic outputs for "effective risk management"  and "cost-sensitive decision-making". However, this practical benefit is never demonstrated. There are no downstream experiments (e.g., using the probabilistic rewards in an RLAIF pipeline) to show that these calibrated probabilities provide a tangible advantage over simpler discrete judges, which could potentially be calibrated post-hoc.
5.  **Missing Robustness Details:** The RL method relies on a parser $g$ to extract the numeric probability from the model's text output. The paper states that the reward function drives the parsability rate $s_{\theta}(x)$ "toward 1"  and that the optimal policy is "parsable everywhere", but it never reports the *empirical* parsability rate. This makes it difficult to assess the robustness of this method during training and inference.

**Questions:**

The main concerns are presented in the weaknesses. Moreover, I am very curious about the exact output distribution of the calibrated probabilities. From my experience, even with calibration, the model would still output some pseudo-probabilities - they do not have mathematical meaning and are basically some sparsed, discontinuous scores like 0.7, 0.4, or something alike.  I wonder what your model's outputs look like, and if it still looks like that, I highly doubt whether the model is just imitating the behavior of outputting scores or it really knows how to predict meaningful probabilities as claimed.

---

> ### Author Response · Authors · 2025-11-24
> **Response by Authors**
>
> Thank you for your constructive and detailed feedback\! We are encouraged that you find our motivation clear and important, and our training methods principled and theoretically sound. We are also glad that you appreciate our rigorous analysis of positional bias and the practical insights on data efficiency. We address your comments in detail below.
>
> >Arbitrary "Ground Truth" Distribution: The entire framework's success is benchmarked against a "ground truth" preference distribution . However, this distribution is synthetically constructed using a single teacher model (Gemini-2.5-Flash) prompted with a specific, ad-hoc set of 7 personas and hard-coded weights (e.g., "Everyday User (4x Weight)" ) at a high temperature (). This target distribution seems highly design-dependent and arbitrary. It is unclear whether this target represents a meaningful, general-purpose population preference or merely a specific artifact of this particular prompt-engineering process.
>
> We acknowledge that the specific target distribution constructed via Gemini-2.5-flash represents one of many possible ground truths. However, we emphasize that our framework is a general-purpose method designed to calibrate to **any** given preference distribution, whether synthetic or human-generated.
>
> To empirically prove that our method is not an artifact of the specific Gemini-based pipeline, we conducted an additional experiment using **GPT-5-mini** as the source for generating preference labels. We finetuned Qwen-2.5-7B using SFT to match this new, distinct target distribution. As shown in the table below, our method achieves nearly identical performance gains and calibration quality regardless of whether the source is Gemini or GPT-5. This lends strong evidence that our approach is agnostic to the source of the preference distribution.
>
> | Preference Source | Alignment (MSE) ↓ | Performance (Agr.) ↑ | Performance (F1) ↑ | Calibration (ECE) ↓ | Calibration (Brier) ↓ |
> | :---- | :---- | :---- | :---- | :---- | :---- |
> | Gemini-2.5-flash | 0.1143 | 0.8264 | 0.5590 | 0.1341 | 0.1394 |
> | GPT-5-mini | 0.1156 | 0.8435 | 0.5961 | 0.1309 | 0.1321 |
>
> Furthermore, while the reviewer questions the arbitrariness of the target, we argue that any preference signal contains inherent uncertainty that standard methods discard. To demonstrate this, we compared our distributional SFT against a standard SFT baseline on the exact same target data.
>
> The results below show that even if we assume the target is a specific artifact, the standard baseline fails to model it accurately, suffering from high calibration error (Expected Calibration Error, or ECE, of 0.1766 vs 0.1341) and severe positional bias (swap consistency of 0.42 vs 0.82). This demonstrates that the improvement is not due to the specific choice of distribution, but rather the modeling paradigm itself.
>
> | Label Type | Alignment (MSE) ↓ | Performance (Agr.) ↑ | Calibration (ECE) ↓ | Consistency ↑ | Abs. Dev. ↓ |
> | :---- | :---- | :---- | :---- | :---- | :---- |
> | Discrete | 0.1462 | 0.7993 | 0.1766 | 0.4252 | 0.5528 |
> | Distribution | 0.1143 | 0.8264 | 0.1341 | 0.8284 | 0.1827 |
>
> >Ignores Preference Structure and Transitivity: The paper models each response pair  as an independent Bernoulli trial. While the authors present this as an advantage that avoids the Bradley-Terry (BT) assumption, it also discards a massive amount of structural information. This formulation does not enforce or model transitivity (i.e., if  and , then ). This structure is fundamental to preference learning and is exploited by standard reward models to improve sample efficiency and logical consistency. The paper provides no analysis of its model's transitivity violations nor any comparison against standard BT-style reward model baselines.
>
> As the reviewer notes, BT models assume a rigid transitivity structure. However, prior studies (e.g.,\[1\]\[2\]\[3\]) have demonstrated that complex preference data often exhibit significant intransitivity. In collective human judgment, this is known as Condorcet’s paradox, where valid population-level disagreement can lead to non-transitive cycles. In such cases, enforcing transitivity forces the model to hallucinate a strict ranking where none exists, discarding valid signals of ambiguity. Our goal is to faithfully report the full preference distribution. By modeling pairs independently, our framework captures these natural intransitivities, whereas a standard BT model would fail to represent them.
>
> \[1\] Chen, Shuo, and Thorsten Joachims. "Modeling intransitivity in matchup and comparison data." Proceedings of the ninth acm international conference on web search and data mining. 2016\.
> \[2\] Munos, Rémi, et al. "Nash learning from human feedback." Forty-first International Conference on Machine Learning. 2024\.
> \[3\] Zhang, Yifan, et al. "Beyond bradley-terry models: A general preference model for language model alignment." arXiv preprint arXiv:2410.02197 (2024).

---

> > ### Author Response · Authors · 2025-11-24
> > **Response by Authors (cont.)**
> >
> > >Training and Evaluation Data Overlap: The main results for alignment (MSE) and calibration (ECE) in Table 1 are reported on a test set where the labels were generated by the exact same Gemini-2.5-Flash \+ persona methodology used to create the training data. While OOD results are provided later, these headline results may primarily demonstrate successful imitation of the teacher's specific (and arbitrary) label distribution, rather than a more general improvement in judgment or calibration.
> >
> > To fully address the reviewer’s concern on generalizability beyond the training distribution, we conduct additional OOD transfer experiments, where we take models finetuned using RL with Brier score on Gemini-2.5-flash annotated calibration data and evaluate them on GPT-5-mini annotated test data.
> >
> > As shown in the table below, our models demonstrate remarkable robustness to this distribution shift. For instance, Qwen-2.5-7B maintains an agreement of **83.6%** on the unseen GPT-5 data (only slightly reduced from 85.8% on the native Gemini data). Similarly, the alignment (based on MSE) and calibration (based on ECE) metrics remain strong. This lends strong evidence that our method learns generalized preference modeling capabilities that transfer across different judges, rather than merely imitating a specific teacher's label distribution.
> >
> > | Model | Transfer? | Alignment (MSE) ↓ | Performance (Agr.) ↑ | Performance (F1) ↑ | Calibration (ECE) ↓ | Calibration (Brier) ↓ |
> > | :---- | :---- | :---- | :---- | :---- | :---- | :---- |
> > | Gemma-2-9B | Yes | 0.1023 | 0.8495 | 0.5980 | 0.1154 | 0.1219 |
> > | Gemma-2-9B | No | 0.0764 | 0.8706 | 0.5895 | 0.0879 | 0.0946 |
> > | Qwen-2.5-7B | Yes | 0.1224 | 0.8355 | 0.5908 | 0.1319 | 0.1407 |
> > | Qwen-2.5-7B | No | 0.0893 | 0.8575 | 0.5804 | 0.1015 | 0.1103 |
> >
> > >No Downstream Task Validation: The paper strongly motivates the need for probabilistic outputs for "effective risk management" and "cost-sensitive decision-making". However, this practical benefit is never demonstrated. There are no downstream experiments (e.g., using the probabilistic rewards in an RLAIF pipeline) to show that these calibrated probabilities provide a tangible advantage over simpler discrete judges, which could potentially be calibrated post-hoc.
> >
> > We clarify that the primary downstream application of an autorater is evaluation (i.e., accurately estimating win rates). In **Appendix D.1**, we explicitly validated this utility by using our probabilistic judge to compute win rates for the Chatbot Arena Conversations benchmark. We observe that our finetuned autoraters can more accurately reflect the true human win rate.
> >
> > Regarding the suggestion to use simpler discrete judges with post-hoc calibration, we emphasize that **Table 1** already includes such comparisons against post-hoc calibration baselines. Our results show that end-to-end distributional training is fundamentally superior to post-hoc calibration. As standard training on discrete labels discards uncertainty information during training, post-hoc methods (like temperature scaling (TS)) can adjust the global confidence, but they cannot recover the fine-grained, instance-level uncertainty that our method learns directly from the preference distribution. This is evidenced by our method's significantly lower ECE (0.1015) and Brier score (0.1103) compared to the TS baseline (0.1529 and 0.1646).
> >
> > >Missing Robustness Details: The RL method relies on a parser  to extract the numeric probability from the model's text output. The paper states that the reward function drives the parsability rate  "toward 1" and that the optimal policy is "parsable everywhere", but it never reports the empirical parsability rate. This makes it difficult to assess the robustness of this method during training and inference.
> >
> > Thanks for the suggestion. We measured the parsability rate on the test set for all finetuned models. The empirical parsability rate for all RL-tuned models is 100%, and our SFT finetuned models also achieve near-perfect parsability (100% for Gemma-2-9B and 99.8% for Qwen-2.5-7B). We will include these statistics in the final revision to explicitly demonstrate the method's robustness.

---

> > > ### Author Response · Authors · 2025-11-24
> > > **Response by Authors (cont.)**
> > >
> > > >The main concerns are presented in the weaknesses. Moreover, I am very curious about the exact output distribution of the calibrated probabilities. From my experience, even with calibration, the model would still output some pseudo-probabilities \- they do not have mathematical meaning and are basically some sparsed, discontinuous scores like 0.7, 0.4, or something alike. I wonder what your model's outputs look like, and if it still looks like that, I highly doubt whether the model is just imitating the behavior of outputting scores or it really knows how to predict meaningful probabilities as claimed.
> > >
> > > We quantitatively verified that these verbalized scores are meaningful using metrics such as ECE and Brier score in our main evaluation. If the model were simply “imitating the behavior of outputting scores”, its calibration error would be high. However, as shown in Table 1, our method achieves a remarkably low ECE and Brier score compared to baselines, confirming that outputs are semantically grounded in the actual preference distribution.

---

> > > > ### Comment · Reviewer_ZXxy · 2025-11-25
> > > >
> > > > Thank the authors for their detailed response and clarifications. I appreciate the explanations regarding the ground truth construction (Point 1), data overlap (Point 3), and parsability robustness (Point 5), which address my initial concerns.
> > > >
> > > > However, I remain unsatisfied regarding Point 2 (Transitivity/BT Assumption) and Point 4 (Downstream Task Validation). As said, the lack of BT-based model baselines greatly undermines the actual value of the proposed method. For the preference structure and transitivity, the current theoretical stance does not justify the exclusion of a BT-based formulation. More importantly, the proposed downstream task is simply another form of evaluation. It is entirely possible to apply your calibration framework to a BT model, but no evidence or empirical rationale is given to show the advantage of the proposed method over BT-based ones. This hinders the practical application value of this work.

---

> > > > > ### Author Response · Authors · 2025-11-26
> > > > >
> > > > > Thank you for your continued engagement. To fully address this concern, we conducted additional experiments by training a reward model head using the standard Bradley-Terry formulation ($p(y^+ \\succ y^- \\mid x) \= \\sigma(r(x, y^+) \- r(x, y^-))$) with a negative log-likelihood objective. We compared this baseline with our RL method trained on the same 50k data with binary pairwise preference labels.
> > > > >
> > > > > As shown in the table below, our method consistently outperforms the BT baseline. This advantage is particularly significant on the Qwen-2.5-7B model, where our method improves alignment (MSE) by $\\sim0.09$, boosts performance (Agreement) by $0.26$, and reduces calibration error (Brier score) by over $50\\%$. We believe these empirical results strongly justify the use of our calibration framework over standard BT-based formulations.
> > > > >
> > > > > | Base Model | Method | Alignment (MSE) ↓ | Performance (Agr.) ↑ | Calibration (Brier) ↓ |
> > > > > | :---- | :---- | :---- | :---- | :---- |
> > > > > | Gemma-2-9B | BT | 0.0956 | 0.7813 | 0.1379 |
> > > > > | Gemma-2-9B | Ours | 0.0764 | 0.8706 | 0.0946 |
> > > > > | Qwen-2.5-7B | BT | 0.1813 | 0.5967 | 0.2336 |
> > > > > | Qwen-2.5-7B | Ours | 0.0893 | 0.8575 | 0.1103 |

---

> > > > > > ### Comment · Reviewer_ZXxy · 2025-11-27
> > > > > >
> > > > > > Thank the authors for their comprehensive rebuttal. Overall, the method has illustrated great potential, but I am still concerned with its practicality, as people mainly rely on LLM-as-a-judge/"autoraters" for their great generalization ability on unseen tasks. The cost of collecting domain-specific preference data and doing reasoning inference is non-negligible, thus hindering its wider applications. With most of the concerns addressed, I will raise my rating accordingly.

---

> > > > > > > ### Author Response · Authors · 2025-11-28
> > > > > > >
> > > > > > > The authors sincerely thank the reviewer for their valuable comments and constructive discussion. We are glad that our response addressed your primary concerns. We will ensure the updated version includes the new results and a detailed discussion on generalization across preference distributions.

---

### Official Review · Reviewer_eNX3 · 2025-10-24

**Soundness:** 4
**Presentation:** 3
**Contribution:** 3
**Rating:** 6
**Confidence:** 3

**Summary:**

This work 1/ argues for the superiority of a probabilistic autorater (LLM used a rating judge), ie a model able to model a whole distribution instead of a single label, to better represent the views of a population ; 2/ suggests two routes to build such distributional autoraters, SFT in the dense case, when multiple annotations are available, and RL in the sparse case, with only one label ; 3/ elects to train models to better verbalize probability, building on the ability of the model to generate natural language rationales ; 4/ shows good results on alignment, calibration as well as resilience to positional biases, with the "RL‑Brier" variant showing the best overall trade‑off ; and 5/ finds their approach to adapt well OOD, to human annotations and to an objective benchmark.

**Strengths:**

- the problem formulation is clear and indeed compelling: estimating probabilities by a model-judge is a nice way to retain the inherent uncertainty from a population ;
- using a verbalized probability is appropriate, principled and well implemented ;
- good experimental results, alignment and calibration, for the two models (Gemma-2-9B and Qwen-2.5-7B) ;
- the distribution-matching fine-tuning appears to eliminate position bias (by looking at what happens when swapping the answers) ;
- the results are confirmed OOD, either with human-annotations or for objective tasks ;
- the extra study on the budget is interesting: it's better to do RL on many prompts annotated once than SFT on less prompts with less annotations.

**Weaknesses:**

- the metrics used for the evaluation rely on the majority label when the whole point is to better model the distribution of views ;
- the distribution for the main experiment comes from Gemini not a human population. That there is a confirmation of the results on the PandaLM dataset is nice but in that case the majority vote is treated as the ground truth ;
- relying on natural language is relevant but it comes with potential issues when parsing to extract the probabilities ;
- the budget study, admittedly not core to this work (but it is presented as "key finding"), feels a bit rushed: one would like to see what happens when using the same prompts for example ;

**Questions:**

- have you looked carefully at the influence of the prompts, and their diversity, on the quality of the modeling?
- similarly, one has to wonder about the magic number of 50K annotations for the training (50x10 or 50x1): have you experimented with less/more annotations?

---

> ### Author Response · Authors · 2025-11-24
> **Response by Authors**
>
> Thank you for your insightful feedback\! We are glad that you found our problem formulation compelling and our method appropriate and principled. It is also encouraging to see that our results regarding calibration, position bias elimination, and the annotation budget trade-offs resonated with you. We provide detailed responses to your questions below.
>
> >the metrics used for the evaluation rely on the majority label when the whole point is to better model the distribution of views ;
>
> We would like to clarify that our evaluation does not rely solely on the majority label. As our goal is indeed to model the full distribution of views, we explicitly used mean squared error (MSE) to measure the alignment (or deviation) between our model's predicted probability distribution and the true preference distribution. While we include standard accuracy metrics on benchmarks like PandaLM and JudgeBench, where the ground truth is defined by the majority label, our primary experimental results demonstrate that our method successfully calibrates autoraters to the underlying preference distribution.
>
> >the distribution for the main experiment comes from Gemini not a human population. That there is a confirmation of the results on the PandaLM dataset is nice but in that case the majority vote is treated as the ground truth ;
>
> Thank you for this comment. The primary goal of our work is to introduce a generalizable framework capable of calibrating autoraters to **any** given preference distribution, regardless of its source. While we utilized Gemini-based distributions as our primary testbed (a standard practice in LLM judge research due to the high cost of large-scale human annotation), our method is not tied to this specific data source. To demonstrate this generalizability, we conducted an additional experiment using SFT to finetune Qwen-2.5-7B on a preference distribution generated by **GPT-5-mini**. We observed improvements in calibration and alignment comparable to those achieved with the Gemini-2.5-flash generated distribution, confirming that our framework successfully calibrates to the underlying target distribution regardless of the source.
>
> | Preference Source | Alignment (MSE) ↓ | Performance (Agr.) ↑ | Performance (F1) ↑ | Calibration (ECE) ↓ | Calibration (Brier) ↓ |
> | :---- | :---- | :---- | :---- | :---- | :---- |
> | Gemini-2.5-flash | 0.1143 | 0.8264 | 0.5590 | 0.1341 | 0.1394 |
> | GPT-5-mini | 0.1156 | 0.8435 | 0.5961 | 0.1309 | 0.1321 |
>
> >relying on natural language is relevant but it comes with potential issues when parsing to extract the probabilities ;
>
> We prioritized this design choice over others (e.g., using a dedicated regression head) because it preserves the model’s versatility. This is accomplished by allowing the same model to output both a calibrated probability and free-form natural language reasoning in a single pass. The result is significantly more interpretable and user-friendly for downstream applications.
>
> Furthermore, our experiments show that this potential issue is effectively mitigated by the finetuning process. We observed that the model learns to adhere strictly to the specified format, achieving a parsability rate of \>99.8% on the test set after finetuning.
>
> >the budget study, admittedly not core to this work (but it is presented as "key finding"), feels a bit rushed: one would like to see what happens when using the same prompts for example ;
>
> In our original design, we deliberately maintained a fixed total budget to explore the trade-off between dataset diversity and label density. However, to directly address your concern, we conducted a new controlled experiment using a fixed set of 5k prompts (randomly sampled from the larger pool) to isolate the impact of the label schema. We compared finetuning Qwen-2.5-7B on this identical set of prompts using SFT with two different targets: 1\) standard discrete labels versus 2\) distribution labels. We observe that the autorater finetuned with distribution labels significantly outperforms the discrete baseline in alignment, calibration, and consistency. Notably, it is far more resilient to positional bias (as measured by swap consistency and symmetric deviation), highlighting the importance of the distribution-matching objective.
>
> | Label Type | Alignment (MSE) ↓ | Performance (Agr.) ↑ | Calibration (ECE) ↓ | Consistency ↑ | Abs. Dev. ↓ |
> | :---- | :---- | :---- | :---- | :---- | :---- |
> | Discrete | 0.1462 | 0.7993 | 0.1766 | 0.4252 | 0.5528 |
> | Distribution | 0.1143 | 0.8264 | 0.1341 | 0.8284 | 0.1827 |

---

> > ### Author Response · Authors · 2025-11-24
> > **Response by Authors (cont.)**
> >
> > >have you looked carefully at the influence of the prompts, and their diversity, on the quality of the modeling?
> >
> > To explicitly verify the influence of prompt diversity versus the modeling objective, we conducted a controlled study (detailed in our response to the comment above). In this experiment, we fixed the set of prompts to an identical 5k sample for both the baseline and our method, thereby eliminating prompt diversity as a variable. The results confirm that the improvement is robust and driven by the distributional matching approach itself, rather than being an artifact of the prompts or their diversity.
> >
> > >similarly, one has to wonder about the magic number of 50K annotations for the training (50x10 or 50x1): have you experimented with less/more annotations?
> >
> > We empirically selected the budget of 50k annotations to strike a balance between performance and computational efficiency. To validate this, we compared the model trained on the reduced set of 5k annotations against our main 50k model. We observed that reducing the annotation budget causes a significant drop in performance. While we observed that scaling beyond 50k yields diminishing returns relative to the increased training and annotation cost, we leave the detailed study of the scaling law as future work.

---

> ### Comment · Reviewer_eNX3 · 2025-11-25
>
> thanks for your responses. I have also read the other reviews and I am leaning towards maintaining my current (favorable) rating.

---

> > ### Author Response · Authors · 2025-11-28
> >
> > The authors sincerely thank the reviewer for their valuable comments and constructive discussion.  We are glad that our response addressed your concerns. We will update our paper to incorporate the comments and discussions.

---

### Official Review · Reviewer_RzUv · 2025-10-30

**Soundness:** 3
**Presentation:** 3
**Contribution:** 2
**Rating:** 2
**Confidence:** 3

**Summary:**

The paper proposes a methodology for training large language models (LLMs) to serve as automated judges. Specifically, the model is trained to predict dense, probabilistic labels through chain-of-thought (CoT) reasoning by using supervised fine tuning (SFT), and subsequently refined using reinforcement learning based on binary labels to further enhance its judgment capability.

**Strengths:**

1. The paper suggests a new reward modeling approach for RL training in LLM. Brier Reward, which differs from the conventional logarithmic reward commonly used for training within the [0, 1] reward range. Which demonstrates high performance compare with previous approach.
2. The paper show that the model trained using this method performs well not only on difference test datasets but also on different objective test set too. This result suggests that the proposed approach generalizes effectively and does not overfit to a specific dataset.
3. The paper is well-written and easy to follow, clearly explaining both the experimental design and the intended objectives behind each component of the study.

**Weaknesses:**

1. Since the proposed method assumes that distributional labels are already provided at the annotation stage, the paper’s contribution is  limited to training a model to accurately predict the given scalar label values. The learning approach extends conventional LLM-as-a-judge tasks, which predict scalar scores, into a continuous (float-level) prediction, which represents only a marginal conceptual advancement.
2. The model is trained through labeled data, yet the primary baselines used for comparison are zero-shot and calibration-based methods that do not involve any training. This makes the comparison inappropriate. It would be more valid to compare against other fine tuning approaches using similar training setups.
3. The paper compares its model against other fine-tuned models as baselines (JudgeLM-7B, PandaLM-7B, etc); however, this comparison is still unfair. A valid evaluation should involve models that share the same training data and base model, differing only in the fine-tuning method. As it stands, the reported results measure absolute performance levels rather than truly assessing the validity or effectiveness of the proposed method.

**Questions:**

1. The paper argues that predicting probabilistic labels rather than binary labels is important for improving model performance. However, generating probabilistic labels inherently requires more data than binary annotations. The paper does not address this issue. It would be interesting to see an experiment where the total number of preference samples is fixed, directly comparing the performance between models trained with binary labels and those trained with probabilistic labels.

2. Does the paper's approach have any explicit advantages compared to "generative reward models" paper?

---

> ### Author Response · Authors · 2025-11-24
> **Response by Authors**
>
> Thank you for your valuable feedback and suggestions\! We are encouraged that you find our paper well written, and recognize the effectiveness of our approach compared to conventional methods and our model's strong generalization across diverse datasets. We provide a discussion of your comments below.
>
> >Since the proposed method assumes that distributional labels are already provided at the annotation stage, the paper’s contribution is limited to training a model to accurately predict the given scalar label values. The learning approach extends conventional LLM-as-a-judge tasks, which predict scalar scores, into a continuous (float-level) prediction, which represents only a marginal conceptual advancement.
>
> We believe there is a key misunderstanding regarding our assumptions about the data. To clarify, we presented two methods (SFT and RL) for calibrating autoraters to a preference distribution tailored towards different data conditions. While our SFT method utilizes continuous labels, our proposed RL method does **not** assume that distributional or continuous labels are provided at the annotation stage.
>
> Instead, a core contribution of our paper is the ability to calibrate autoraters to full preference distributions without access to continuous labels. We achieve this by utilizing standard binary labels via reinforcement learning with proper scoring rules such as Brier and Log scores. To the best of our knowledge, we are the first to introduce a general probabilistic framework to calibrate autoraters to model the full preference distribution by recovering calibrated continuous uncertainty estimates from discrete preference data.
>
> >The model is trained through labeled data, yet the primary baselines used for comparison are zero-shot and calibration-based methods that do not involve any training. This makes the comparison inappropriate. It would be more valid to compare against other fine tuning approaches using similar training setups.
>
> Since we are the first to propose calibrating autoraters to preference distributions, there is no existing finetuning method that targets this specific objective. We did, however, compare our model against other finetuned judges, specifically JudgeLM and PandaLM. Yet it should be recognized that these models are trained to predict a single deterministic label, whereas our goal is to recover the full preference distribution. Due to the fact that no existing finetuning methods share this distributional objective, we relied upon zero-shot/calibration baselines as the primary comparison for **calibration** metrics. We have also conducted a new controlled experiment using the requested training setup, where our method outperforms the standard SFT across all performance, calibration, and bias metrics. Please see our response below for the detailed results.

---

> > ### Author Response · Authors · 2025-11-24
> > **Response by Authors (cont.)**
> >
> > >The paper compares its model against other fine-tuned models as baselines (JudgeLM-7B, PandaLM-7B, etc); however, this comparison is still unfair. A valid evaluation should involve models that share the same training data and base model, differing only in the fine-tuning method. As it stands, the reported results measure absolute performance levels rather than truly assessing the validity or effectiveness of the proposed method.
> >
> > To fully address the reviewer's concern about training setups, we conducted a new experiment where we finetuned the same base model (Qwen-2.5-7B) on the same training data using the conventional standard SFT approach (training on single ground truth labels). Our distributional SFT method significantly outperformed this controlled baseline on both calibration and performance metrics. Moreover, our method is less susceptible to positional bias, as demonstrated by the higher swap consistency and lower symmetry deviation. This lends strong evidence that the distributional learning objective is crucial for the performance gain.
> >
> > | Method | Alignment (MSE) ↓ | Performance (Agr.) ↑ | Performance (F1) ↑ | Calibration (ECE) ↓ | Calibration (Brier) ↓ |
> > | :---- | :---- | :---- | :---- | :---- | :---- |
> > | Baseline | 0.1462 | 0.7993 | 0.5404 | 0.1766 | 0.1766 |
> > | Ours | 0.1143 | 0.8264 | 0.5590 | 0.1341 | 0.1394 |
> >
> > | Method | Consistency ↑ | Abs. Dev. ↓ |
> > | :---- | :---- | :---- |
> > | Baseline | 0.4252 | 0.5528 |
> > | Ours | 0.8284 | 0.1827 |
> >
> > >The paper argues that predicting probabilistic labels rather than binary labels is important for improving model performance. However, generating probabilistic labels inherently requires more data than binary annotations. The paper does not address this issue. It would be interesting to see an experiment where the total number of preference samples is fixed, directly comparing the performance between models trained with binary labels and those trained with probabilistic labels.
> >
> > Thank you for this insightful suggestion. We actually did investigate this exact trade-off in our paper. In Table 1, we compared the data efficiency of learning from probabilistic versus binary labels under a fixed annotation budget. Specifically, we compared two strategies (namely, SFT with dense probabilistic labels and RL with sparse binary labels) with the same total number of preference samples.
> >
> > Our results show that, when the total annotation budget is fixed, our RL method trained on sparse binary labels significantly outperforms the SFT model trained on dense probabilistic labels. This is a central finding of our work, which demonstrates that we do not need expensive probabilistic annotations to achieve good calibration. Instead, our RL approach can effectively recover the full preference distribution from cheaper standard binary feedback.
> >
> > >Does the paper's approach have any explicit advantages compared to "generative reward models" paper?
> >
> > Our work addresses a fundamentally different challenge. The Generative Reward Models (GenRM) work \[1\] uses Chain-of-Thought to improve judgment accuracy on complex reasoning tasks such as math and code. By contrast, our work focuses on calibrating autoraters to capture the inherent subjectivity and uncertainty of human preferences. Specifically, our approach offers several notable advantages. First, unlike GenRM, which forces a single correct answer, our method explicitly models preference distributions, preserving minority viewpoints and ambiguity (aleatoric uncertainty) that is essential for subjective tasks. Second, our approach outputs a calibrated score in a single forward pass, avoiding the high computational cost of GenRM’s 32-sample majority voting. Third, we directly minimize calibration error using proper scoring rules, whereas GenRM optimizes for reasoning accuracy without guaranteeing calibrated confidence scores.
> >
> > \[1\] Mahan, Dakota, et al. "Generative reward models." arXiv preprint arXiv:2410.12832 (2024).

---

> > > ### Comment · Reviewer_RzUv · 2025-11-28
> > >
> > > Thank you for your response. It seems there was a slight misunderstanding on regarding the main contributions of the paper, and most of the concerns addressed. I would like to raise my score to 6, but it appears that the system is currently not allowing modifications. I will update my score once this is resolved.

---

> > > > ### Author Response · Authors · 2025-11-28
> > > >
> > > > The authors sincerely thank the reviewer for their valuable comments and constructive discussion.  We are glad that our response has been helpful in clarifying the misunderstanding and addressing your concerns. We will update our paper to incorporate these comments and discussions.

---

### Official Review · Reviewer_9YcT · 2025-11-01

**Soundness:** 3
**Presentation:** 3
**Contribution:** 2
**Rating:** 4
**Confidence:** 3

**Summary:**

This paper proposes a method for calibrating LLM-as-a-judge (that is, autoraters) for preference distributions. More specifically, the authors go beyond the binary label and model the whole preference distribution as a Bernoulli distribution of which the output is the original binary label. The discussions are made within verbalized probabilities. The authors suggest that the Bernoulli's $p$ can be learned and calibrated by probabilistic labels, including either averaging $m$ binary labels (and then do direct SFT) or adopting a loss between binary labels and probabilistic outputs (and then do RL). The methods show improvements over alignment MSE, preference accuracies, and calibration performance when compared to methods from zero-shot baselines and calibration baselines.

**Strengths:**

1. This paper is clearly written. Going beyond binary labels is an active research trend. This paper motivates why single binary labels are insufficient. The paper is also well-organized and easy to follow.

2. The model is simple but effective. The authors model the learning problem as learning the Bernoulli's $p$, which is one step beyond binary labels. Although I think $p$ itself might not be anough (for example, if $p = 0.5$, there are differences between what's controversial (e.g., two groups of people favoring strongly for different answers) and what's moderately vague (e.g., all people do not show strong preference between different answers)), I do think $p$ is good point to start.

3. The empirical results are valid. I think the authors have included the necessary details of empirical results.

**Weaknesses:**

1. The source of label itself. The label comes from a "teacher" model (also known as the oracle). I agree that it's hard to collect real human-annotated data under probabilistic labels so LLM-as-a-judge can work as a proxy. But the problem is, the performance is also evaluated with that "teacher" model's probabilistic outputs. The effectiveness of the method is kind of a circular argument: we can improve LLM-as-a-judge by calibrating it with another perfectly calibrated LLM-as-a-judge (the teacher model). I'm not criticizing using LLM-as-a-judge as a proxy for human-annotated data, but assuming the access to oracle (no matter human or LLM) label distribution is a bit unrealistic.

2. The OOD performance is not satisfying. As I have discussed in the previous point, the main results are made under the assumption that the training data and the evaluation come from the same source. It means that the learner has access to (a noisy version of) the oracel $p^\ast$; Using that additional information in the training unsurprisingly increases the performance. But for other cases where the OOD issue is possible, the performance is not satisfying (see Table 3 and Table 4).

**Questions:**

1. Please identify my concerns about the source of label. What are the outcomes if you adopt real human-annotated data as both the training and the evaluation data, instead of LLM-as-a-judge? I think that's important. Maybe the real human-annotated data is too limited to do such training, but you can provide results when training data is from "LLM-as-a-judge trained from human data" and evaluation is conducted on real human data (no OOD).

2. Please identify my concerns about the OOD issue. I think it's very unsurprising that you get improvement with access to the oracle $p^\ast$ and train your model on that. But what if you only get one LLM-as-a-judge as the training data and another as the evaluation data? I would appreciate it if you capture your calibration error/performance drop within the scope of OOD severity.

---

> ### Author Response · Authors · 2025-11-24
> **Response by Authors**
>
> Thank you for your thoughtful feedback\! We are glad that you find our paper clearly written, our method simple but effective, and our empirical results valid. We address your comments in detail below.
>
> >The source of label itself. The label comes from a "teacher" model (also known as the oracle). I agree that it's hard to collect real human-annotated data under probabilistic labels so LLM-as-a-judge can work as a proxy. But the problem is, the performance is also evaluated with that "teacher" model's probabilistic outputs. The effectiveness of the method is kind of a circular argument: we can improve LLM-as-a-judge by calibrating it with another perfectly calibrated LLM-as-a-judge (the teacher model). I'm not criticizing using LLM-as-a-judge as a proxy for human-annotated data, but assuming the access to oracle (no matter human or LLM) label distribution is a bit unrealistic.
>
> Thanks for the thoughtful comment. We would like to clarify two critical points:
>
> (1) Our method does **not** require access to the full oracle label distribution during training. As is typical in machine learning, we require only discrete preference pairs sampled from the underlying distribution. The goal is to infer the latent probability from these discrete samples. This is a realistic setting applicable to any data source, whether it be a judge model (like Gemini/GPT) or aggregated human annotations (crowdsourcing).
>
> (2) Our method is a general framework that allows for calibrating autoraters to **any** given preference distributions. To demonstrate this, we conducted a new experiment using GPT-5-mini as a completely different source for ground truth. We finetuned Qwen-2.5-7B using SFT to calibrate to this new, independent target distribution. As shown below, our method successfully learns to model the uncertainty of GPT-5 just as well as it did for Gemini-2.5-Flash. This confirms that our approach is agnostic to the preference source.
>
> | Preference Source | Alignment (MSE) ↓ | Performance (Agr.) ↑ | Performance (F1) ↑ | Calibration (ECE) ↓ | Calibration (Brier) ↓ |
> | :---- | :---- | :---- | :---- | :---- | :---- |
> | Gemini-2.5-flash | 0.1143 | 0.8264 | 0.5590 | 0.1341 | 0.1394 |
> | GPT-5-mini | 0.1156 | 0.8435 | 0.5961 | 0.1309 | 0.1321 |

---

> > ### Author Response · Authors · 2025-11-24
> > **Response by Authors (cont.)**
> >
> > >The OOD performance is not satisfying. As I have discussed in the previous point, the main results are made under the assumption that the training data and the evaluation come from the same source. It means that the learner has access to (a noisy version of) the oracle; Using that additional information in the training unsurprisingly increases the performance. But for other cases where the OOD issue is possible, the performance is not satisfying (see Table 3 and Table 4).
> > >Please identify my concerns about the OOD issue. I think it's very unsurprising that you get improvement with access to the oracle and train your model on that. But what if you only get one LLM-as-a-judge as the training data and another as the evaluation data? I would appreciate it if you capture your calibration error/performance drop within the scope of OOD severity.
> >
> > To directly address the reviewer’s concern about OOD generalization, we performed an additional transferability experiment on our RL-tuned models where the training data (Gemini-2.5-Flash) and evaluation data (GPT-5-mini) originated from completely different sources. To quantify the OOD severity, we measure the disagreement between the training preference source (Gemini) and the evaluation preference source (GPT-5). They disagree on 38.52% of the test items, representing a substantial distributional shift. Despite this, our finetuned autoraters demonstrated remarkable resilience. As shown in the table below, while there was a natural reduction in the metrics compared to the in-distribution setting, the performance remains high. For example, Qwen-2.5-7B maintains an agreement of **83.55%** on the unseen GPT-5 distribution (a drop of only \~2.2% from the in-distribution baseline). This provides strong evidence that our method learns generalizable preference features rather than merely memorizing the specific artifacts from the training distribution.
> >
> > | Model | Transfer? | Alignment (MSE) ↓ | Performance (Agr.) ↑ | Performance (F1) ↑ | Calibration (ECE) ↓ | Calibration (Brier) ↓ |
> > | :---- | :---- | :---- | :---- | :---- | :---- | :---- |
> > | Gemma-2-9B | Yes | 0.1023 | 0.8495 | 0.5980 | 0.1154 | 0.1219 |
> > | Gemma-2-9B | No | 0.0764 | 0.8706 | 0.5895 | 0.0879 | 0.0946 |
> > | Qwen-2.5-7B | Yes | 0.1224 | 0.8355 | 0.5908 | 0.1319 | 0.1407 |
> > | Qwen-2.5-7B | No | 0.0893 | 0.8575 | 0.5804 | 0.1015 | 0.1103 |

---

> > > ### Author Response · Authors · 2025-11-24
> > > **Response by Authors (cont.)**
> > >
> > > > Please identify my concerns about the source of label. What are the outcomes if you adopt real human-annotated data as both the training and the evaluation data, instead of LLM-as-a-judge? I think that's important. Maybe the real human-annotated data is too limited to do such training, but you can provide results when training data is from "LLM-as-a-judge trained from human data" and evaluation is conducted on real human data (no OOD).
> > >
> > > We have indeed included such results where the model is finetuned on Gemini-annotated calibration datasets and tested on the human-annotated PandaLM testset (Table 3), where our method achieves higher agreement and F1 than the previous SOTA model PandaLM, which is finetuned on prompts from the same data distribution (i.e, PandaLM training set).

---

> > > > ### Comment · Reviewer_9YcT · 2025-11-26
> > > >
> > > > I thank the authors for their experiments on the OOD issue. Although the experiments are not thorough due to time limit, I think it at least shows that the method does not fully rely on the assumption that the evaluation model is identical to the training data source. In that light, I'm not against seeing it accepted (I improved my rating accordingly). However, a ``discrete'' oracle is still an oracle, thus I still find the significance limited and the result unsurprising to some level, so my rating will not be higher than threshold.

---

> > > > > ### Author Response · Authors · 2025-11-28
> > > > >
> > > > > The authors sincerely thank the reviewer for their valuable comments and constructive discussion. We are glad that our response has been helpful in addressing your concerns. We acknowledge your remaining note regarding the discrete oracle. We agree that learning a distribution from discrete samples is well-grounded in standard learning theory. However, we believe demonstrating that this calibration holds robustly for modern autoraters (even under distribution shifts) is a valuable empirical confirmation for the community. We will ensure the updated version includes the new OOD results and a detailed discussion on the training data source to reflect these limitations and contributions accurately.

---

### Author Response · Authors · 2025-12-01
**Revision Uploaded**

We thank all reviewers again for the constructive discussion. We have uploaded a revised manuscript to incorporate the additional empirical results and clarifications promised during the rebuttal, with key changes highlighted in blue. We believe these additions help further strengthen the paper's contribution and robustness.

Best regards,

The Authors

---

### Meta-Review · Area_Chair_P2fp · 2026-01-05

**Summary:**

The authors argue that LLM autoraters should model the full distribution of human preferences (rather than providing discrete preference labels) and propose a general framework to calibrate probabilistic autoraters to a target population. Two learning methods are introduced: supervised fine-tuning for dense probabilistic labels and reinforcement learning for sparse binary feedback. Experiments show that distribution-matching training improves calibration, reduces positional bias, and better aligns probabilistic judgments with human preferences while maintaining performance on objective tasks.

Originally, the reviewers were a bit sceptical and raised a couple of concerns, e.g., regarding the setting and issues related to the idea of autorating, the experimental setup and performance, etc. The authors could clarify most issues during the discussion, and in the end, all reviewers are on the acceptance side, albeit only marginally. Indeed, none of them is really enthusiastic about the paper.

I had a look at the paper myself, and in my opinion, it remains below the standards expected for this conference. It's a decent paper, but I would say that the authors are really overselling their contribution, both in terms of novelty (probability estimation and uncertainty quantification have been intensely studied for LLMs in the last couple of years) and technical contribution, which is rather modest. Essentially it boils down to estimating the parameter of a Bernoulli distribution. Why do we need a proof that the mean of a sequence of m i.i.d. outcomes is an unbiased estimate, and that variance decreases as 1/m? With all due respect, that's material of a first semester statistics course. Likewise, it is well known that strictly proper scoring rules (such as Brier, log-loss) incentivise unbiased point estimates.

**Reviewer Concerns:**

The reviewer concerns have basically been addressed. However, as I said, I don't think the paper meets the standards of the conference.

**Reviewer Scores:**

I would not have expected further changes.

---

### Decision · Program_Chairs · 2026-01-26

Reject